# MOCKINGBIRD: PLATFORM FOR ADAPTING LLMS TO GENERAL MACHINE LEARNING TASKS

## ABSTRACT

Large language models (LLMs) are now being used with increasing frequency as chat bots, tasked with the summarizing information or generating text and code in accordance with user instructions. The rapid increase in reasoning capabilities and inference speed of LLMs has revealed their remarkable potential for applications extending beyond the domain of chat bots. However, there is a paucity of research exploring the integration of LLMs into a broader range of intelligent software systems. In this research, we propose a paradigm for leveraging LLMs as *mock functions* to adapt LLMs to general machine learning tasks. Furthermore, we present an implementation of this paradigm, entitled the *Mockingbird* platform. In this paradigm, users define *mock functions* which are defined solely by method signature and documentation. Unlike LLM-based code completion tools, this platform does not generate code at compile time; instead, it instructs the LLM to role-play these *mock functions* at runtime. Based on the feedback from users or error from software systems, this platform will instruct the LLM to conduct chains of thoughts to reflect on its previous output, thereby enabling it to perform reinforcement learning. This paradigm fully exploits the intrinsic knowledge and in-context learning ability of LLMs. In comparison to conventional machine learning methods, following distinctive advantages are offered: (a) Its intrinsic knowledge enables it to perform well in a wide range of zero-shot scenarios. (b) Its flexibility allows it to adapt to random increases or decreases of data fields. (c) It can utilize tools and extract information from sources that are inaccessible to conventional machine learning methods, such as the Internet. Finally, we evaluated its performance and demonstrated the previously mentioned benefits using several datasets from Kaggle. Our results indicate that *Mockingbird* is acceptably competitive for use in real-world applications.

## 1 INTRODUCTION

Currently, large language models (LLM) are widely used in intelligent systems as a chat bot to assist users in summarizing, processing, and generating text. However, this only exploits the natural language processing capabilities of LLMs; there are many other capabilities are not fully utilized, such as the significantly improved reasoning capabilities of recent LLMs (Valmeekam et al., 2024).

A natural idea is to extend LLMs to non-linguistic tasks. Currently, a popular solution is to use LLMs as code generators to generate code for machine learning pipelines (Hollmann et al. (2023), Guo et al. (2024)); this type of method can significantly enhance automated machine learning workflows, but the static nature of pipeline code limits the use of the dynamic capabilities of LLMs. In 2020, Brown et al. (2020) has found that language models are able to learn from their context, proving the basic feasibility of this idea; since then, many researchers have studied the properties of this ability in specific domains: Kirsch et al. (2022); Chan et al. (2022); Jin et al. (2023); Bigelow et al. (2024) Kossen et al. (2024); these work have well explained the nature of in-context learning ability, but provide little overall insight into the integration of LLMs as run-time components into automatic intelligent systems with a wider range of tasks. In order to fill this paucity and to encourage more domains to benefit from the progress of LLM, we propose *Mockingbird*, a versatile and controllable paradigm for adapting LLMs to general machine tasks, and a corresponding open-source platform implementing this paradigm.

Inspired by the finding that the intelligence of LLMs is merely role-playing (Shanahan et al., 2023), we assume that LLMs can also acquire good performance in role-playing functions, and design our paradigm upon this assumption. The fundamental component of *Mockingbird* is *mock function*, a specific type of functions that do not have method bodies, but only have function declaration including documentation and method signatures (contracts on the function parameters and return values). In this paradigm, users are able to use *mock functions* as if they were ordinary functions with function bodies. In contrast to other LLM-drive code generators, *Mockingbird* does not implement these *mock functions* with code generated by LLMs at compile time; instead, it instructs LLMs to 'role-play' them at runtime.

To elaborate further, the platform furnishes LLMs with program metadata including the method signature and accompanying documentation, retrieved from the program; then, function calls to *mock functions* are redirected to LLMs; parameters are packed into user messages, and return values are unpacked from assistant messages. In this manner, LLMs are employed as general-purpose functions. By leveraging the in-context learning capability of LLMs, users can influence the behavior of these *mock functions* with minimal effort by modifying the input-output message pairs presented within the chat histories of LLMs. Furthermore, we present several optional techniques that enhance practicality of this paradigm. For instance, we introduce the substitution script acceleration, which replaces LLM-driven executors with substitution-scripts generated by LLMs, thereby reducing the time consumption significantly. Figure 1 shows a high-level overview of this paradigm.

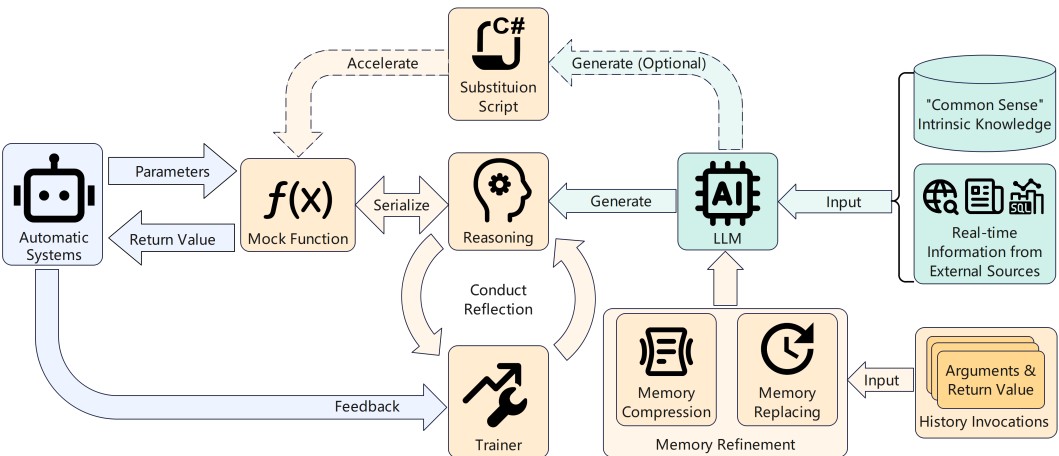

Figure 1: High-level overview of *Mockingbird*. Mock functions redirect ordinary function calls to the LLM, instructing the LLM to initiate reasoning and then generate the return value. Mock trainers use feedback to instruct the LLM to conduct reflection process on its previous errors. Optional modules such as substitution script, memory compression and memory replacing are introduced to further improve the practicality of this paradigm.

This paradigm offers an intuitive interface for systems that have difficulty adapting to chat-driven execution mode, thereby enabling them to leverage the capabilities of LLMs.

This work investigates the potential of applying this paradigm to general machine learning tasks. Furthermore, we implement an extensible machine learning framework for *mock functions*, which is capable of automatically conducting reflection on the incorrect output during the training process. In comparison to conventional machine learning methods, such as those based on statistics or artificial neural networks, *Mockingbird* has the following distinctive advantages (a) the intrinsic knowledge of LLMs acquired from the pre-training data enables this paradigm to perform well on few-shot and zero-shot tasks; (b) this paradigm is not constrained by a strict input data schema, allowing it to process incomplete data entries with missing fields. (c) this paradigm is readily capable of utilizing tools and extracting information from non-structural sources, which are typically inaccessible to conventional machine learning techniques. The aforementioned advantages can serve to enhance the robustness and flexibility of automatic intelligent systems.

We evaluate the general applicability of this paradigm across a range of machine learning tasks from Kaggle. Overall, it achieves acceptably competitive scores compared to conventional machine learning methods, even outperforming many human competitors on several datasets.

## 2 MOCKINGBIRD

This paradigm is composed of following components:

**Mock Function**  A mock function is a function that is defined solely in terms of its method signature (types and other restrictions on parameters and return value), with optional documentation provided to describe its purpose and any remarks pertinent to its use, and it is role-played by LLMs. The fundamental responsibilities of *mock functions* can be summarized as follows: (a) providing LLMs with metadata to role-play the function; (b) handling conversions between program objects and chat messages in JSON format; (c) guaranteeing the formal correctness of responses generated by LLMs through the validation of JSON schema. Figure 2 illustrates the fundamental workflow of a mock function. Mock functions can be employed directly without training process; however, in such instances, it is advisable to provide comprehensive annotation regarding the anticipated behavior of these functions as documentation comments within the code itself.

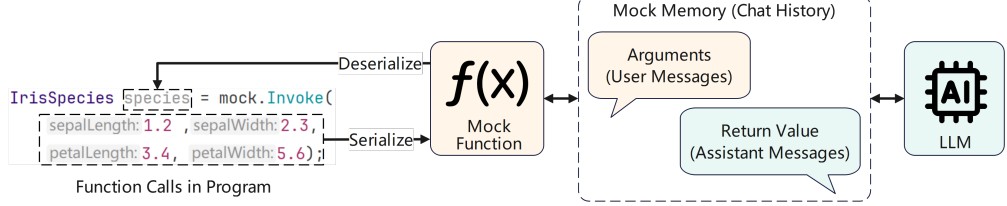

Figure 2: Basic workflow of a mock function. Mock functions automatically handle the conversion between program objects and chat messages, thus communicating between the program and LLMs.

**Mock Invocation**  Mock invocations are designed to conveniently update history invocations in the context. They represent a mapping of invocations in the program space to request-response message pairs in the chat history. Mock invocations automatically update the contents of request and response messages in the chat history when the their parameters or return values are changed. They serves as a simple interface for editing the context.

**Mock Memory**  Mock memories are enhanced chat histories for this paradigm, whose elements are mock invocations instead of chat messages. In addition to invocation management, a branch control feature has been implemented for mock memories. More information about the chat history and the mock memory can be found in the appendix A.1.

**Mock Trainer**  The introduction of mock trainers serves to streamline the process of training and evaluating mock functions, while also assisting users of conventional machine learning frameworks in swiftly adapting to our paradigm. In the training stage, if LLMs output incorrect answers in a mock invocation, a reflection procedure will be conducted by the mock trainer to avoid making similar mistakes in future. The reflection procedure will be covered in detail in section 2.3.

**Substitution script**  A substitution script is a script generated by LLM to represent the behavior learned from the invocation history. This technique is introduced as an optional feature to reduce the time consumption, though this may result in a compromise of accuracy. Details can be found in appendix A.4.

### 2.1 WORKFLOW

The following list delineates the machine learning workflow with *Mockingbird*:

1) **Setup mock function.** According to the metadata including method signature, documentation, and type information of parameters and return value, the mock function generate the system prompt consisting of directives and JSON schemas separately for parameters and return value.

This prompt not only instructs the LLM to give a return value that matches the JSON schema, but also commands it to output the reasoning in the "remarks" field before outputting the return value in the "results" field. Mock functions add this system prompt into the message history during the setup phase. Details are discussed in section 2.2.

2) **Prepare serializers.** According to the type information collected in step 1, the platform generates and code of JSON serializers for parameters and return value, which are then loaded subsequent to their compilation.

3) **Perform an invocation with a LLM.** Once the parameters have been serialized into JSON data that adheres to the schema built in step 1, the JSON data should be appended to the message history as a user message. The message history should then be submitted to the LLM for a response. Upon receipt of the assistant message, mock functions decode return value from the JSON data contained within the response. Then a mock invocation is constructed and registered with the user message and the corresponding assistant message.

4) **Conduct reflection procedure.** In the training phase, the mock trainer compares the return value yielded by the assistant with the ground truth in the training data. If the difference exceeds the threshold set by the users, the mock trainer initiates a reflection procedure. Parameters, return value, and the ground truth are appended to a sub-branch of the main memory branch. The LLM is then instructed to reason why such mistakes are made and to summarize notes on how to avoid making similar mistakes in the future. These notes are called "reflection notes" in this paradigm. Details are discussed in section 2.3.

5) **Update invocations in the mock memory.** Once the reflection notes have been obtained, mock invocations containing incorrect return values are updated. Their "results" fields are amended to the ground truth, while their "remarks" fields are replaced with reflection notes.

6) **Refine the mock memory.** The context length is typically smaller than the size of the training dataset, which means that not all data entries of the training dataset can be directly stored in the context. When the context of a mock function reaches the limitation of length, the mock trainer initiates a memory refinement procedure. We provide a memory replacing policy and a memory compression policy (see section 2.4 for details) along with our implementation of this paradigm. Mock trainers are designed to be highly customizable, allowing users to configure them with their own memory refinement policies.

7) **Generate substitution script.** When the substitution script technique is enabled, then the mock script component instructs LLMs to generate script code for the role-played function based on the information within the context. Its script code is not immutable, and once the behavior of a mock function changes (due to reflection or manual instruction), the current substitution script is invalidated and this generation process starts again. Details are described in the appendix A.4.

## 2.2 GUARANTEE OF FORMAL CORRECTNESS

Figure 3 shows the process of building system prompt for mock functions. The platform retrieves the documentation for the delegate (function declaration) from the program documentation file generated by the compiler, and extracts the semantic meanings for this delegate, parameters and return value. After that, JSON schemas for parameters and return values are generated separately with the type information acquired from the runtime. Semantic meanings are added into "description" fields of the corresponding elements in schemas. All these schemas are defined in the standard format of JSON schema, rather than expressed in natural languages.

When instructions on the format of responses are described in natural languages, there is a possibility of a mismatch between the format of the responses and the expected format due to the ambiguity in the instructions. Even if instructions are clear without ambiguity, LLMs may still fail to obey these instructions due to the overlay of reasoning patterns (Jiang et al., 2024a). Solving this formal randomness is at vital importance: in contrast to user-oriented applications, formal correctness of responses are crucial for automatic systems. The return value cannot be correctly found and parsed in one request-response round if it is stored in field with random names. Furthermore, even if automatic systems discover an error with the schema, re-generation of responses will significantly increase the time consumption. The importance of JSON schema for input data is discussed in appendix A.2.

A number of studies have put forward the idea of fine-tuning-based solutions as a means of improving the ability of LLMs to follow instructions (Wallace et al. (2024);Jiang et al. (2024a)). But with an extra emphasis on general availability, we prefer to rely on non-fine-tuning solutions. Recent

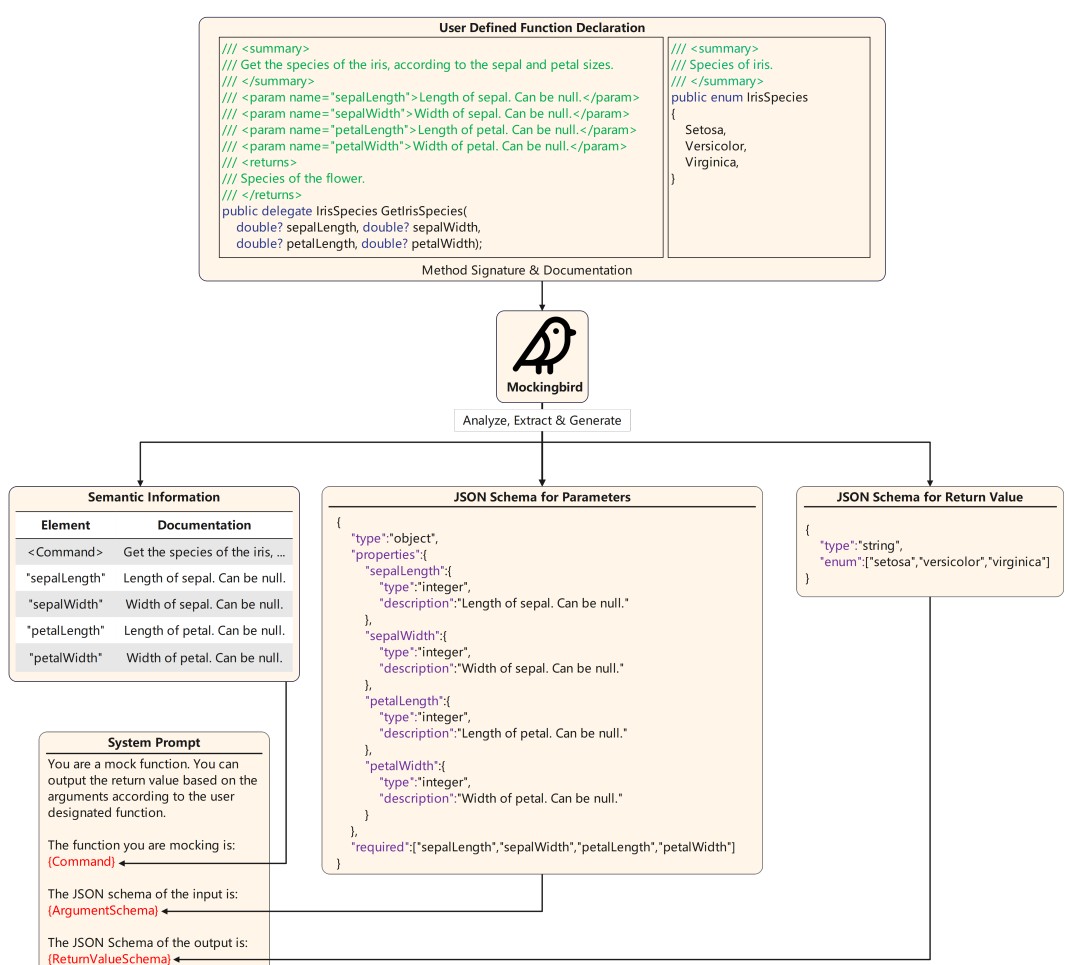

Figure 3: The system prompt for mock functions are built according to the semantic information and JSON schema for parameters and return value. Semantic information explains the semantic meanings of parameters, which is crucial for LLMs to understand the task and parameters. JSON schemas for parameters and return values are used to constrain the layout and semantic meanings of data fields, ensuring the correct mutual understanding of data fields between program and LLMs.

LLMs, such as *GPT-4o* has provided *structural output* feature that strictly limits the schema of the output. *Mockingbird* fully exploit this feature to guarantee the formal correctness of responses.

For LLMs which currently do not support this feature, mock functions have a embedded JSON schema validator to verify the formal correctness of responses, and reject illegal responses with detailed error report to initiate another request-response round. However, this re-generation process will introduce additional time-consumption. And during our experiment on models with various numbers of parameters (appendix B), formal errors are common to observe in LLMs with relatively small numbers of parameters, therefore, we highly recommend users fine-tune LLMs with relatively small numbers of parameters to schema-based structured outputs.

## 2.3 LEARNING THROUGH REFLECTIONS

Even though our experiments show that the intrinsic knowledge LLMs acquired from pre-training enables them to reach relatively high scores on multiple tasks with zero-shot, the capacity for continuous improvement still remains a crucial consideration in practical applications.

Inspired by the workflow of neural network based machine learning frameworks, we introduce a reflection mechanism into *Mockingbird*. Figure 4 shows the workflow of reflection and the prompt used by mock trainers to instruct the LLM for reflection to perform reflections.

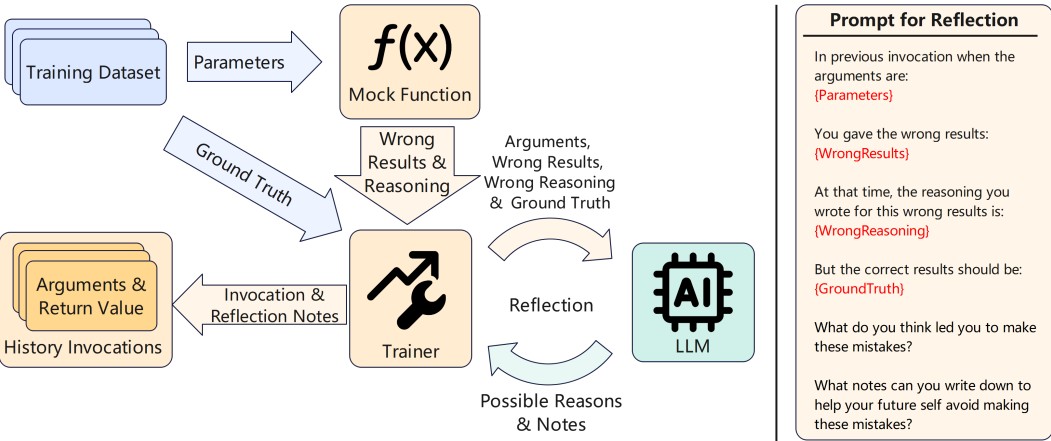

Figure 4: **Left:** Workflow for reflection procedure. After acquiring the wrong results from the mock function, the mock trainer will instruct the LLM to reflect on the possible reasons for making such mistakes and summarize notes for not making similar mistakes in future invocations. **Right:** Prompt used by mock trainers to conduct reflections.

We implement mock trainers to automatically host and manage the learning procedure. In a training session, the mock trainer will iterate every data entry in the dataset, feeding parameters into the mock function, comparing the results obtained from the mock function with the ground truth. If the difference between the output results and ground truth exceeds the threshold preset by users, then a reflection procedure will be conducted: the mock trainer will generate a reflection instruction including the wrong results, wrong reasoning and the ground truth; thereafter, the LLM will be instructed to analyze the possible reasons for making these mistakes, and summarize notes to avoid making similar mistakes. The response from the LLM, consisting of possible reasons and notes for avoiding similar mistakes, is called "reflection notes" in this paradigm. Then, results field of the invocation to reflect will be corrected to the ground truth, and the reasoning field will be replaced with reflection notes.

In contrast to previous methods of in-context learning which focus more on the patterns of examples in the context (Liu et al. (2022);Bhattamishra et al. (2024)), we expect LLMs to reflect on their previous reasoning flows and learn by adapting to reasoning flows which are more likely to be correct, rather than relying solely on the in-context learning ability of LLMs.

## 2.4 MEMORY REPLACING AND COMPRESSION

Most of current LLM inference services are stateless, which means that history messages must be submitted again with the latest request message in subsequent calls. Due to concerns for token consumption, this leads to the urgent need for strategies to reduce the number of context tokens while minimizing the damage to the information within the context.

Many researches have proposed strategies for selecting examples (Zhang et al. (2022); Zhang et al. (2023); Lee et al. (2023); Qin et al. (2024); Nguyen & Wong (2023); Peng et al. (2024); S. et al. (2024)). These studies all show competitive improvement on specific domains, so from the perspective of paradigm design, instead of select one or more selection strategy, we implement the mock trainer in a highly customizable design. We provide a default implementation of the replacing policy by replacing correct invocations (without reflection notes) with the latest reflected invocations.

As for the context compression, current methods focus more on the token level (Liskavets et al. (2024);Ge et al. (2024)) and even on the level of modifying the underlying implementation of transformers (Yu et al. (2024)). According to their evaluation results, these methods are effective enough, but they are too heavy for *Mockingbird*, since this paradigm is designed to work simultaneously with

different LLMs at various price tiers. In addition, there is an undeniable difficulty in using these solutions with commercial close-source LLMs for the time being. Similar to the methods proposed by Wang et al. (2024) and Jiang et al. (2024b), we provide a default implementation of semantic compression ("soft compression"), by instructing LLMs to summarize the reflection notes to compress previous invocations.

Details of the implementation for default memory replacing and memory compression algorithms are in appendix A.3.

# 3 EVALUATION

We evaluate the performance of *Mockingbird* on several general machine learning tasks, covering the most common categories of classification and regression. It must be emphasized that our aim is to provide a more suitable paradigm for exploiting the capabilities of LLMs in automatic intelligent systems, rather than provide a *state-of-the-art* method that is comprehensively superior to conventional machine learning methods.

The performance evaluation is conducted to prove the effectiveness of this paradigm, and therefore it is included in the main text. The scalability evaluation on different LLMs with various scales of parameters and corresponding discussion is in appendix B. The performance evaluation with retrieval-augmented generation is appendix D.

## 3.1 PERFORMANCE EVALUATION

In this section, we evaluate its performance separately on classification and regression datasets acquired from Kaggle. The goal of this evaluation is to: (a) explore the possible special properties of this platform as an unconventional machine learning method; (b) evaluate its usefulness for non-expert users. Therefore, its performance is assessed by comparing with scores of human competitors in corresponding Kaggle leaderboard[1], which can be considered as an estimated upper bound of the scores that non-professional users can achieve using conventional machine learning methods.

Table 1 shows its performance on classification tasks, and table 2 is for performance on regression tasks. Context length is the maximum number of invocations that the mock function can memorize through the training process; a context length of 0 means that the training process is skipped, which could represent the extent to which users can treat this paradigm as an "out-of-the-box" solution. Both evaluations are performed with *GPT-4o* as the underlying LLM.

We selected these datasets based on the following criteria: (a) there is at least one competition for this dataset on Kaggle; (b) enough human competitor teams participate that competition; (c) the leaderboard is available for public access.

Table 1: Accuracy evaluation on classification tasks, compared with scores of human competitors from Kaggle leaderboard. Datasets are: *Titanic Survival Prediction* (Cukierski, 2012), *Poisonous Mushrooms Classification* (UCI, 1981), *Horse Colic* (McLeish & Cecile, 1989) and *Insurance Cross-Selling*. **Humans' Best** is the best scores achieved by human competitors in the leaderboard; and **Outperformed** is the proportion of human competitors whose scores are outperformed by the best scores using this paradigm. The best accuracy is underlined.

| Dataset | Accuracy↑ with Context Length | | | | | Humans' Best | Outperformed |
|---|---|---|---|---|---|---|---|
| | 0 | 20 | 40 | 60 | 80 | | |
| Titanic | 0.7879 | 0.7887 | 0.7743 | 0.7918 | 0.7866 | 1.0000 | 92.29% |
| Mushrooms | 0.3720 | 0.5693 | 0.7302 | 0.9968 | 0.8359 | 0.9851 | 100% |
| Horses | 0.5450 | 0.5531 | 0.5836 | 0.5532 | 0.5087 | 0.7818 | 6.42% |
| Insurances | 0.5880 | 0.6730 | 0.7260 | 0.6989 | 0.6793 | 0.8975 | 18.35% |

---

[1]Raw leaderboard data can be found in supplementary materials. Data of leaderboard is updated to Oct 1st, 2024.

Table 2: Root mean square error (RMSE) and median absolute error (MedAE) evaluation on regression tasks, compared with scores of human competitors from Kaggle leaderboard. Datasets are: *Used Car Price Prediction*, and *Prediction of Mohs Hardness*. **Humans' Best** is the best scores achieved by human competitors in the leaderboard; and **Outperformed** is the proportion of human competitors whose scores are outperformed by the best scores using this paradigm. The best scores are underlined.

| Dataset | Error↓ with Context Length | | | | | Humans' Best | Outperformed |
|---|---|---|---|---|---|---|---|
| | 0 | 20 | 40 | 60 | 80 | | |
| Car Prices | 22225 | 22523 | 28915 | 24534 | 22794 | 62917 | 100% |
| Mohs Hardness (MedAE)[†] | 0.6000 | 0.5800 | 0.6000 | -[‡] | -[‡] | 0.2500 | 60.86% |
| Mohs Hardness (RMSE) | 1.2803 | 1.1596 | 1.1314 | -[‡] | -[‡] | -[†] | -[†] |

[†] The leaderboard on Kaggle uses median absolute error to rank human competitors.
[‡] This dataset only contains 57 data entries.

## 3.2 DISCUSSION

**Longer context is not all you need.** Among these evaluations, we find that best performances are not guaranteed to be accompanied with the biggest context length. For instance, on *Poisonous Mushroom Classification* task, the performance peak appears around a context length of 40, and the performance decreases by 16.14% when the context length grows to 80. There is an extreme example on *Mohs Hardness Prediction* task. When the context length is 40, while 70.18% of the data entries are reflected and then stored in the context, the RMSE only reduces by 2.43%. Increasing context length is not a silver bullet that can magically improve performance in any kind of tasks. This phenomenon is not as incomprehensible as it seems to be. Kossen et al. (2024) has pointed out that LLMs do not treat all data within the context equally, but tends to rely on examples semantically closer to the queries; also, they cannot fully resist their preference acquired through the pre-training, which is also shown in the performance plateau on *Titanic Survival Prediction Dataset*. In other words, intrinsic knowledge and the reasoning ability have made LLMs more than in-context learning machines; this leads us to reconsider the characteristics of LLMs machine learning, rather than continuing to view it simply as combining in-context learning of transformers with intrinsic knowledge.

**Intrinsic knowledge does not always help.** On *Poisonous Mushroom Classification* task, we observed a strange initial accuracy bellowing random guessing: 0.3720, which is 25.6% worse than random guessing. In this configuration, the context length is zero, which means there is no history invocations or reflection notes stored in the context which could influence its behavior, therefore its judgment is solely relying on its intrinsic knowledge. If we humans know the fact that our accuracy is significantly bellowing 0.5, then by simply reversing our final answers, we can reach a relatively high accuracy on such binary classification tasks. However, even when LLMs know they are outputting the wrong answer, they still need a sufficient examples and reflections in learning process to reduce the influence of their "harmful" intrinsic knowledge; in this task, that is the process of accuracy increasing from 0.3720, going through 0.5693, 0.7302, and finally reaches a high score outperforming the best score of human competitors. Additional discussion with detailed examples is in appendix E.

**Intrinsic Knowledge Cannot Make Up for the Lack of Domain-specific Knowledge.** The performance of LLMs on *Horses Health Outcome Prediction* is especially bad. Similar to their performances on *Insurance Cross-selling Prediction*, they starts with an accuracy of nearly random guessing. Their accuracy improves as the size of training dataset grows on *Insurance Prediction* task until reaches the context length of 40, however, on *Horses Health Prediction* task, such improvement is not observed. This phenomenon indicates that the reflection process is barely not effective in this task. After investigating the operation logs, we found that this is very likely caused by the lack of details in the reflection note. For example, one of reflection notes contains a tip, "*Ensure a comprehensive overview is taken, recognizing how severe signs work together to indicate decline,*

*leading to death, thus helping differentiate from intervention-based outcomes.*". This tip contains no factual errors, but important details such as how to achieve the goal of "ensuring a comprehensive overview is taken" and "recognizing how severe signs work together" is not written. Therefore, this tip is more like a directional requirement rather than an actionable error-correction solution. One natural speculation for the reason is that the intrinsic knowledge of LLMs does not contains such detailed domain-specific knowledge, thus LLMs can only make reflection notes of such directional requirements based on its common sense within the intrinsic knowledge. Huang et al. (2024a) has reported that current LLMs cannot discover the errors within their reasoning process in the absence of external feedback, meanwhile, the only external feedback in the training stage is the ground truth from the dataset, which can only indicates the existence of errors but cannot identify the errors. If the intrinsic knowledge cannot cover this specific domain, then LLMs may struggle to identify and correct errors within their previous reasoning process. This suggests that documents containing domain-specific knowledge provided through retrieval-augmented generation or human feedback in the training stage is probably still needed. Additional discussion can be found in appendix F.

## 4 RELATED WORK

This work relies on LLMs' semantic understanding and in-context learning ability to learn the relationship between the history input and output, intrinsic knowledge and reasoning ability to predict the output according to the given input. The training related techniques are engineered on the basis of in-context learning theories. The dynamic characteristics brought by role-playing nature differs it from LLM-based code generation solutions. The core idea of in-context learning that directly considers LLMs as learning machines distinguishes it from current automatic machine learning (AutoML) solutions, whose essence is LLM-based code generation.

**Adapting LLMs to Machine Learning Tasks** Recently, Kashyap & Sinha (2024) also examined the feasibility of integrating LLMs into statistical learning workflows on "Titanic Survival Prediction" dataset; they claimed to reach a "significant improvement" with data preprocessing and thought refinement (chain of thought). However, our experiments show that the scores are relatively higher without data pre-processing and other additional steps introduced in their work, which means that their method is less effective than they claimed to be. Moreover, their method relies heavily on the user's knowledge of statistical learning, which is not friendly to the automatic intelligent systems.

**In-Context Learning** Brown et al. (2020) have discovered that LLMs are able to learn from analogies based on the context, and summarize this ability as "in-context learning". Based on solid experimental evidence, Kossen et al. (2024) draws the following important conclusions about in-context learning: (a) content in context can influence the behavior of LLMs, which means that in-context learning is indeed effective to some extent; (b) examples in the context that are semantically closer to the queries have more effect, and vice versa; this makes in-context learning different from conventional machine learning methods that treat all training data equally; (c) it is impossible to eliminate the preference that LLMs acquired through pre-training by in-context learning, but this preference can be reduced to some extent by prompting. These conclusions are consistent with our findings in experiments.

**Code Generation and Self-Repair** The delayed progressive generation of substitution scripts distinguished it from other code generation solutions, for it does not rely on self-repair, but incorporates information from the parameter-result pairs and corrects its behavior through reflections with external feedback. Olausson et al. (2023) have shown that LLMs struggle to repair the errors on their own, but that the effectiveness of repairs improves significantly when human feedback is provided. Also, Huang et al. (2024a) have demonstrated that the reasoning performance cannot be improved without external feedback. Our design of deferring the substitution script generation is consistent with these findings. Meanwhile, compared to user assisted code generation solutions (Zhu-Tian et al. (2024); Fakhoury et al. (2024)), the feedback of substitution scripts can be provided by software systems rather than solely from humans users, which is more suitable to automatic intelligent systems.

**LLM-Powered Code Generation for Automated Machine Learning Workflow** These researches trying to utilize LLMs to assistant data scientists, by instructing LLMs to automatically generate code for pipelines consisting of conventional machine learning components. Guo et al. (2024) presents a framework that utilize case-based-reasoning LLMs to automatically understand the task and thus generating pipeline code in Python to compose existing conventional machine learning

components. Hollmann et al. (2023) presents a similar framework that also produce pipeline code in Python, with a different goal to incorporate domain-specific knowledge into automated machine learning and accomplish automatic feature engineering. Tang et al. (2024) and Huang et al. (2024b) have designed benchmarks for these code-generation-based automated machine learning workflows. Some readers may confuse researches in this field with our work; actually, we serve different goals with different approaches. These researches are trying to instruct LLMs to write conventional machine learning code to assist data scientists; however, we are trying to provide software developers with a alternative choice when they cannot rely on data scientists nor conventional machine learning methods by instructing LLMs to role-play functions.

## 5 CONCLUSION

In this work, we present the paradigm for adapting LLMs to general machine learning tasks based on *mock functions*, which instruct LLMs to role-play the function defined by users. We also propose a machine learning framework for mock functions, whose usage is similar to that of conventional neural network based machine learning frameworks. Optional techniques, including substitution script generation, memory refinement policies for replacing and compression are also introduced to address its shortcomings in inference cost and time consumption. This paradigm can fully exploit the reasoning ability, in-context learning feature, and intrinsic knowledge of LLMs, together with its dynamic nature, making it an out-of-the-box solution for automatic intelligent systems. Guidelines for deploying *Mockingbird* in resource constrained environments in appendix G.

Finally, we evaluate its performance and scalability on several machine learning tasks from Kaggle, and it shows acceptable results, and even outperforms most the human competitors in several tasks. We then discuss several findings about LLM machine learning based on the analysis of the evaluation results and the operational logs of the platform.

**Limitation and Future Work**

- Currently, it is not financially feasible to evaluate our paradigm on all possible types of datasets on all possible machine learning tasks, and we also have a lack of domain-specific datasets; therefore, we believe that validating this paradigm on a wider range of tasks could provide useful insights into the properties of LLM machine learning.
- *Mockingbird* only provides a complete framework, in which the many implementations of techniques have the potential to be optimized with research results from the field of in-context learning. For example, the memory replacing technique can be optimized with the progress made in example comparing and selecting methods proposed in in-context learning research.
- Some of the current AutoML methods, such as *AutoGluon* (Erickson et al., 2020), can reach state-of-the-art results on many Kaggle datasets. There is a possibility for *Mockingbird* to combine the advantages of LLM machine learning and conventional machine learning methods by wrapping AutoML modules as tools for LLMs to configure and invoke at inference time.
- Our current analysis can only be conducted on the operation logs, which only contains the external information of LLMs (such as the reasoning content and reflection notes in text), with a lack of internal information insides LLMs, such as the neuron activation state. With the internal information of LLMs, there is a chance to calculate and visualize the actual weight of different sentences in the reasoning content and reflection notes, and therefore optimize the reflection mechanism. Such tools may also help users diagnose and correct the reasoning errors of LLMs at run-time.
- If future research can develop a general-purpose system for computing the similarity between different invocation arguments, then there is opportunity to reduce the token consumption by treating history invocations as retrieval-augmented generation materials. Instead of providing all history invocations in the context, searching the vector database for the best matching history invocations will significantly reduce the token consumption, and may also assist LLMs in the reasoning process. Meanwhile, this technique will also enable *Mockingbird* to be used in a reinforcement learning manner.

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

## A  TECHNICAL DETAILS OF THE IMPLEMENTATION

This section describes the technical details of implementing *Mockingbird*.

### A.1  CHAT HISTORY AND MOCK MEMORY

As we have described above, a mock memory is an advanced chat history. So before we describe the details of a mock memory, we need to explain the role of chat histories within the implementation of LLM clients first.

**Chat History**  Readers who are already familiar with the implementation of LLM clients can skip this part. Unlike chat on websites, LLM service APIs are mostly stateless: service providers do not store history messages on servers. Every time developers want LLMs to respond to a new message in an interactive chat, they need to send all of the history messages along with that new message; and then the LLM will go through all these messages again (including messages written by users and those generated by the LLM) to generate a new response message. This list of history messages and this new message waiting for a response is called a "chat history"; and in the field of in-context learning, it is usually referred to as a "context".

A chat history, or a context, is the place where states of a chat session are majorly stored. Rather than a single user message, the message history is the minimum unit of messages to be sent to LLMs. Also, the essence of LLM-powered chat bots can be seen as the completion of these chat histories. The new message from users must be appended to the end of a chat history, otherwise the LLM cannot get information about the chat context. The response messages from LLMs must also be appended to the chat history after request messages from users, otherwise LLM will answer these "unanswered" requests in subsequent API calls. Also, the wrong order of requests and corresponding

responses can cause an abnormal halt in the API call, especially when tool calls or corresponding tool call results are missing or misplaced.

**Mock Memory**   Mock memories are enhanced chat histories for this paradigm, whose elements are mock invocations instead of chat messages. In addition, a branch control feature has been implemented for mock memories. This feature allows users to create sub-branches from a main branch, which will "inherit" the current invocation history of the main branch. Sub-branches can be dropped or committed back to the creation time-point in the main branch. This feature provides isolation between different branches, allowing multiple LLM tasks to run in parallel; otherwise, no further requests can be added to the memory until the response is received, as LLMs may be responding to requests from other tasks. Figure 5 shows the process of creating sub-branches and committing them back to the main branch.

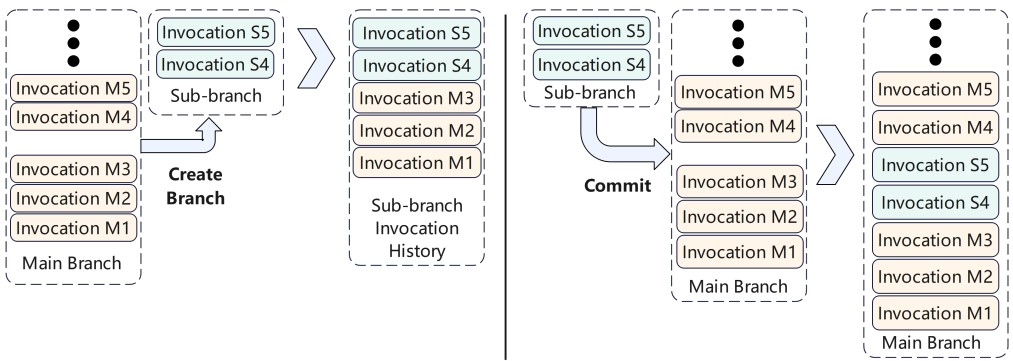

Figure 5: Branch control feature of mock memories. **Left**: when a branch is created, subsequent invocations added to the main branch will not affect the sub-branch. **Right**: when a sub-branch is committed, its invocations are merged into the main branch at the location where the sub-branch was created.

A.2   DISCUSSION ON THE IMPORTANCE OF INPUT SCHEMA

The JSON schema for parameters does not seem to be as important as the schema for the return value in the inference process: the structure of the parameters is self-explanatory in the JSON data. However, if one or more parameters have value ranges (min/max values), or if some of them are enumeration types, additional information about their ranges and all possible values in the JSON schema can help LLMs perform reasoning. In this sense, the lack of schema for parameters does not compromise the stability of the system. However, they are essential for substitution script generation, because LLMs need accurate and detailed information to write the formally correct script. In addition to explicitly explaining the semantic meaning of parameters, JSON schemas for parameters can also indicate the existence of nested objects, and provide additional information about types such as the min/max values, and all possible values for enumeration types.

A.3   MEMORY REPLACING AND COMPRESSION

**Memory Replacing**   When the memory replacing feature is enabled, and the number of history invocations within the context has reached the user-specified limits, the replacing algorithm is executed to drop one history invocation. In the training stage, the reflection procedure is only conducted on invocations that LLMs output incorrect results. Therefore, the history invocations that LLMs output the correct results should be replaced with the latest invocation first; and when there is no such invocation to replace, the earliest invocation in the context will be selected as the replacing candidate to compare with the latest invocation. If the latest invocation has correct results, then it will be dropped; otherwise the earliest invocation will be replaced. Algorithm 1 is the pseudo code for this algorithm.

**Memory Compression**   The default implementation for memory compression relies entirely on the summarizing ability of LLMs. This is done by appending an instruction of summarizing history invocations to the context. History invocations are removed from the context, and then the summary

---

**Algorithm 1:** Default Memory Replacing Algorithm

---

**if** replacing_enabled $\neq$ *true* | history_invocations.*Length* $<$ replacing_threshold **then**
⌊ **return**
**foreach** invocation *in* history_invocations **do**
⎢ **if** invocation.*expected_result* $\neq$ invocation.*actual_result* **then**
⎢ ⎢ invocation $\leftarrow$ new_invocation
⎢ ⌊ **return**

**if** new_invocation.*expected_result* $==$ invocation.*actual_result* **then**
⌊ **return**
history_invocations.Remove$history\_invocations.first$
history_invocations.Append$new\_invocation$
**return**

---

generated by the LLM is inserted into the context. In text 1, the text above the dotted line is the prompt to instruct the LLM to summarize the history invocations; and the text below the dotted line is the message to replace the history invocations.

---

Text 1:   Prompt for Memory Compression

In previous invocations you have made some mistakes and remarks to not make them again. Now summarize these notes to help your future self to avoid these mistakes and maximize your accuracy. You can include specific examples and reasoning in the summary.

- - - - - - - - - - - - - - - - - - - - - - - - - - - - - - - - - - - - - - - - - - - - - - - - - - -

Here are notes summarized by yourself to help you avoid mistakes and maximize accuracy:
**{compressed_notes}**

---

### A.4   SUBSTITUTION SCRIPT

In contrast to other LLM-drive code generators which generate code at the outset, the substitution script is generated only after the LLM has acquired sufficient information about the "correct" behavior of the mock function. This distinguishes it from other compile-time code generators. This lead to an obvious advantage that history invocations and reasoning contents within the mock memory can help LLMs better understand the actual purpose and the mechanism of the function. Although the use of substitution script significantly reduce the time consumption (to constant time level), the effect of substitution script is not always equal to the dynamic reasoning process of LLMs, especially when LLMs fail to express the complicated logic in code. Therefore, we make it an optional technique.

Figure 6 shows the simplified workflow for generating a substitution script. In the *Mockingbird* implementation, the substitution script component manages this entire process automatically. The metadata for the role-played delegate including command, documentation and method signature are provided to the LLM, and then the LLM is instructed to generate the script code based on this information. Once the substitution script has been generated and compiled, subsequent invocations will be redirected to the substitution script instead of the LLM. Note that the content of the substitution script is not final, as the reflection process will invalidate the script, always assuming the behavior of the mock function has been changed by the reflection process.

Since *Mockingbird* is implemented in C#, the compilation process is necessary; for scripting languages such as Python, this process can be skipped. However, LLMs are prone to making mistakes when using some unusual APIs (such as accessing non-existent methods of "BsonDocument"), so this "error & retry" process can expose semantic errors early and thus increase the stability of the intelligent system. Also, it is necessary to explain in addition that the training stage functions a little different when substitution script is enabled: (a) the invocations are redirected to the substitution script, so that the context including history invocations are not used; (b) the substitution script is regenerated only when the LLM produces incorrect results in the training stage, which means that invocations for which the LLM produces correct results do not contribute to the increase in accuracy

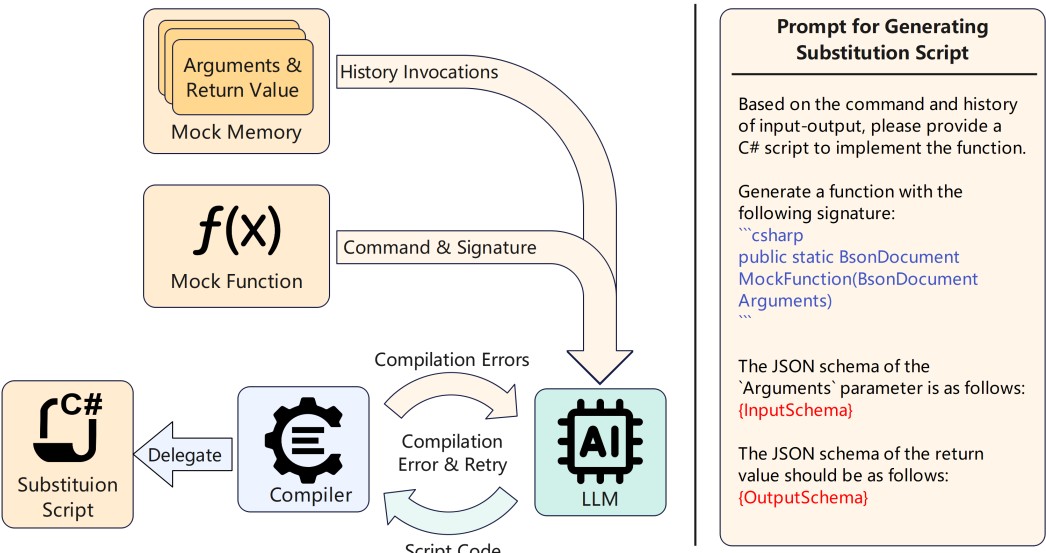

Figure 6: **Left:** Workflow for generating Substitution Script. The LLM is instructed to generate the source code based on the method information and history invocations; the compiler will try to compile the generated code and report the compilation error to the LLM, until the compilation is successful. The source code and corresponding compiled delegate are stored in the Substitution Script component. **Right:** Prompt for the LLM to generate the script code.

if the generation process is not restarted (because the current substitution script continues to produce correct results).

In our experiments with the substitution script component, we have observed varying degrees of accuracy loss in almost all of the the tasks tested. In some tasks (such as "Iris Classification"), surprisingly, LLMs can use a simple composition of if-else sentences to achieve acceptable scores. Details of the substitution script evaluation can be found in the appendix C

For example, code 1 is the substitution script generated by *GPT-4o* for the "Iris Classification Task". This script achieves an accuracy of 92.72%, which is close to the accuracy of 98.18% for real-time reasoning; and the average time consumption per invocation is reduced from 1284.2118ms to 0.0755ms.

# B    SCALABILITY EVALUATION

To assess the scalability of *Mockingbird*, we evaluate its multiple indicators on the "Titanic Survival Prediction" task using several close and open source models with different parameter sizes as underlying LLMs. The context length is set to 40. Optional techniques such as substitution scripts are disabled.

These indicators are: (a) accuracy; (b) time consumption; (c) token consumption; (d) effectiveness of reflection.

These models are: GPT-4o, GPT-4o-Mini, GPT-4-Turbo, GPT-3.5-Turbo; Qwen-2.5 (72B, 32B, 14B, 7B, 3B, 1.5B, 0.5B), Llama-3.2 (11B, 3B, 1B), Llama-3.1 (70B, 8B), Gemma-2-9B and Mistral-7B.

## B.1    ACCURACY AND EFFECTIVENESS OF REFLECTION

In this section, we evaluate the performance of *Mockingbird* using different underlying LLMs. We test its performance with a context length of 40, and compare its accuracy with the zero-shot con-

```
Code 1:   Substitution Script for Iris Classification

using MongoDB.Bson;
public static BsonDocument MockFunction(BsonDocument Arguments)
{
    double? sepalLength = Arguments.Contains("sepalLength") ?
        (double?)Arguments["sepalLength"].AsDouble : null;
    double? sepalWidth = Arguments.Contains("sepalWidth") ?
        (double?)Arguments["sepalWidth"].AsDouble : null;
    double? petalLength = Arguments.Contains("petalLength") ?
        (double?)Arguments["petalLength"].AsDouble : null;
    double? petalWidth = Arguments.Contains("petalWidth") ?
        (double?)Arguments["petalWidth"].AsDouble : null;
    // Default results
    string remarks = "Insufficient data provided.";
    string species = "Unknown";
    bool isReadyToCompile = true;
    // Basic logic to mock species determination based on petal
        length (for simplicity)
    if (petalLength.HasValue)
    {
        if (petalLength < 2.5)
        {
            species = "Setosa";
            remarks = "Petal length indicates Setosa.";
        }
        else if (petalLength < 5.0)
        {
            species = "Versicolor";
            remarks = "Petal length indicates Versicolor.";
        }
        else
        {
            species = "Virginica";
            remarks = "Petal length indicates Virginica.";
        }
    }
    else
    {
        remarks = "Petal length is required to determine the
            species.";
        isReadyToCompile = false;
    }
    // Building the response
    var response = new BsonDocument
    {
        { "Remarks", remarks },
        { "Results", species },
        { "IsReadyToCompile", isReadyToCompile }
    };
    return response;
}
```

figuration (with a context length of 0), to assess the effectiveness of the reflection mechanism when using underlying LLMs with a wider range of parameter sizes.

In this evaluation, the "Formal Correctness Ratio" is the proportion of responses that match the schema and can therefore be successfully and correctly parsed by the software system. This property is important in real-world applications, because lower formal correctness usually leads to more re-generation and longer latency between requests and valid responses.

Table 3: Accuracy evaluation on "Titanic Survival Prediction" task using different LLMs with various parameter sizes as underlying LLMs. This table shows the accuracy of different models with different context lengths. "Formal" represents the ratio of formal correct responses to all responses. "Accuracy Improvement" indicates the improvement of accuracy compared to zero-shot configuration (with a context length of 0).

| Model | Context Length 0 | | Context Length 40 | | Accuracy Improvement | Formal Correctness Improvement |
|---|---|---|---|---|---|---|
| | Accuracy | Formal | Accuracy | Formal | | |
| Gemma-2-9B | 0.7340 | 0.3350 | 0.6897 | 0.8476 | -0.0443 | +0.5126 |
| Mistral-7B | 0.6049 | 0.8583 | 0.6627 | 0.9562 | +0.0578 | +0.0979 |
| Llama-3.1-8B | 0.5768 | 0.8477 | 0.4465 | 0.9681 | -0.1303 | +0.1204 |
| Llama-3.1-70B | 0.7093 | 0.9834 | 0.7614 | 1.0000 | +0.0521 | +0.0166 |
| Llama-3.2-1B | 0.5230 | 0.1917 | 0.5099 | 0.6109 | -0.0131 | +0.4192 |
| Llama-3.2-3B | 0.5061 | 0.8501 | 0.5593 | 0.9649 | +0.0532 | +0.1148 |
| Llama-3.2-11B | 0.6285 | 0.8559 | 0.4312 | 0.9895 | -0.1973 | +0.1336 |
| Qwen-2.5-0.5B | 0.4163 | 0.5303 | 0.4688 | 0.8043 | +0.0525 | +0.2740 |
| Qwen-2.5-1.5B | 0.5679 | 0.9195 | 0.5229 | 0.7086 | -0.0450 | -0.2109 |
| Qwen-2.5-3B | 0.6139 | 0.9748 | 0.6992 | 0.9884 | +0.0853 | +0.0136 |
| Qwen-2.5-7B | 0.6644 | 0.9966 | 0.7137 | 0.9918 | +0.0493 | -0.0048 |
| Qwen-2.5-14B | 0.7766 | 0.8804 | 0.8131 | 0.9942 | +0.0365 | +0.1138 |
| Qwen-2.5-32B | 0.7744 | 0.9988 | 0.8002 | 0.9530 | +0.0258 | -0.0458 |
| Qwen-2.5-72B | 0.7833 | 0.9944 | 0.7802 | 0.9930 | -0.0031 | -0.0014 |
| GPT-3.5-Turbo | 0.7878 | 0.9966 | 0.7755 | 1.00 | -0.0123 | +0.0034 |
| GPT-4-Turbo | 0.7890 | 1.00 | 0.8202 | 0.9977 | +0.0312 | -0.0023 |
| GPT-4o | 0.7833 | -* | 0.8108 | -* | +0.0275 | -* |
| GPT-4o-Mini | 0.7598 | -* | 0.7990 | -* | +0.0392 | -* |

* The structured output feature of *GPT-4o* guarantee the formal correctness of responses.

From the performance results shown in table 3, we can see that the accuracy improvement varies according to the different models, and it is common to see a little performance improvement or even performance degradation for LLMs with small parameter sizes. Regarding the ineffective reasoning and learning for LLMs with small parameter sizes, we perform an analysis on their operation logs, and conclude 3 types of reasons that compromise the effectiveness of the reasoning and learning process: hallucinations, inconsistency, and formalism.

**Hallucinations About Inputs and Facts**   These hallucinations can be categorized as follows:

- Hallucinations about the input parameters, such as falsely claiming that a passenger has siblings on board when the corresponding argument is zero[2], and identifying a 15-year-old passenger as a child[3];
- Hallucinations about the facts, such as an unfounded claim that a passenger has died, "*The passenger was a third-class male with an age of 39; however, he died as the ship sank and lifeboats only accommodated women and children.*"[4], and making claims that contradict the statistical facts, "*Class 3 was the third-class passenger class. Males had a higher survival rate than females in first and second class due to class distinctions.*"[5];

**Inconsistency Between Reasoning and Decision**   The prediction results produced by LLMs with small parameter sizes may be inconsistent with their reasoning content. For example, the LLM generates the reasoning content "The passenger survived."[6](which is also unfounded), but it still predicts the possibility of survival as 0.

**Formalism for Reflection and Learning** There are four types of formalism during the reflection and learning processes:

- Directional requirements that lack of actionable details or facts, such as "*Have a solid understanding of algorithms that predict class survival such as Logistic Regression, [...] I will focus on ensuring correct input handling and proper calculations based on specific formulas that are*

---

[2]Log item 67443dee4a52aee384278fa8, in "Titanic-Qwen2.5-0.5b-Context_0-41.6386%.json"

[3]Log item 674432656a79acc35d9dccc2, in "Titanic-Qwen2.5-0.5b-Context_0-41.6386%.json"

[4]Log item 67442ed63225f8aab39c9491, in "Titanic-Llama3.2-3b-Context_0-50.6173%.json"

[5]Log item 67442ed73225f8aab39c9494, in "Titanic-Llama3.2-3b-Context_0-50.6173%.json"

[6]Log item 67443dea4a52aee384278f9a, in "Titanic-Qwen2.5-0.5b-Context_0-41.6386%.json"

*part of a known algorithm rather than relying solely on general logic or incorrect reasoning.*". This can also be seen in the reflection notes generated by models with large parameter sizes, such as " *Ensure that the reasoning provided aligns with the data and does not overlook significant factors that could influence outcomes. Double-check the model's predictions against known historical trends to catch any discrepancies early on.*", which also lacks detail on how to implement these tips.

- Irrelevant responses, such as "*I'm sorry, but I need more information about what you want me to do next based on the original request. Could you please provide additional context or a clearer description of what actions would like to be taken and why?*"[7], and "*I previously given the wrong result with an argument of [male, 40], therefore it is an invalid argument for an airline. I am also confused now about the correct answer.*"[8](this is a reasoning content rather and not a reflection note). This can also take the form of simply describing the arguments and do not directly elaborating on the reasoning process, such as "*The passenger was a female with one sibling aboard.*"[9].

- Superficial imitating, i.e. the LLM does not understand the meaning of the reflection notes and thus imitates to generate "reflection notes" as the reasoning content. After the reflection procedure, the "remarks" field (the reasoning content) is replaced by the reflection notes, which usually start by admitting of the errors generated by the LLM. During our experiment, we have observed that an LLM imitates to generate "reflection notes" that start with sentences like "*I recognize the mistake in my previous calculations and apologize [...]*"[10] and "After correcting the input data and recalculating based on a known algorithm [...]"[11], even before the reflection procedure begins and even if it outputs the correct results. This behavior is not to use the remarks fields of history invocations as reference material, but to imitate the pattern in the text of these fields.

## B.2 TIME CONSUMPTION

In this section, we analyze the time consumption during the training stage and evaluation stages separately. The GPU environment for local deployment is NVIDIA A6000 Ada $\times 4$ (49,140 MB vRAM for each, 196,560 MB vRAM in total). As the GPU environment for online LLM service providers (for *GPT* series in this evaluation) is unknown and unlikely to be identical to our local deployment environment, this comparison result is for reference only.

Figure 7 shows the box plot of the time consumption during the training stage. In the training stage, reflection process is performed for invocations that the LLM gives the wrong answer, so the accuracy will also affect the average time consumption. It also shows that models with large parameter sizes have a wider range of time consumption. Models with more parameters usually require more inference time per token, meanwhile they have the ability to generate longer texts; these two factors together make them have a much higher upper bound of time consumption.

Figure 8 shows the box plot of the time consumption during the evaluation stage. There is no reflection process in this stage, so the time consumption is obviously lower than in the training stage. The general rule shown in this graph is that models with more parameters usually take more time to process these invocations.

## B.3 TOKEN CONSUMPTION

Table 4 shows the token consumption and corresponding money cost for these models working on the "Titanic Survival Prediction" task with a context length of 40. This dataset contains 891 entries of data. The token price for open source LLMs is referenced from the an online LLM inference service provider *ToghetherAI*[12].

---

[7]Log item 674733c91325795a8cc33831, in "Titanic-Qwen2.5-0.5b-Context_40-46.8860%.json"

[8]Log item 674733d11325795a8cc33836, in "Titanic-Qwen2.5-0.5b-Context_40-46.8860%.json"

[9]Log item 67443dec4a52aee384278fa1, in "Titanic-Qwen2.5-0.5b-Context_0-41.6386%.json"

[10]Log item 674727b1463aaffe10958dd9, in "Titanic-Qwen2.5-1.5b-Context_40-52.2914%.json"

[11]Log item 674727b0463aaffe10958dd8, in "Titanic-Qwen2.5-1.5b-Context_40-52.2914%.json"

[12]The website URL for token price is "https://www.together.ai/pricing". Pricing data is captured on Nov. 25th, 2024.

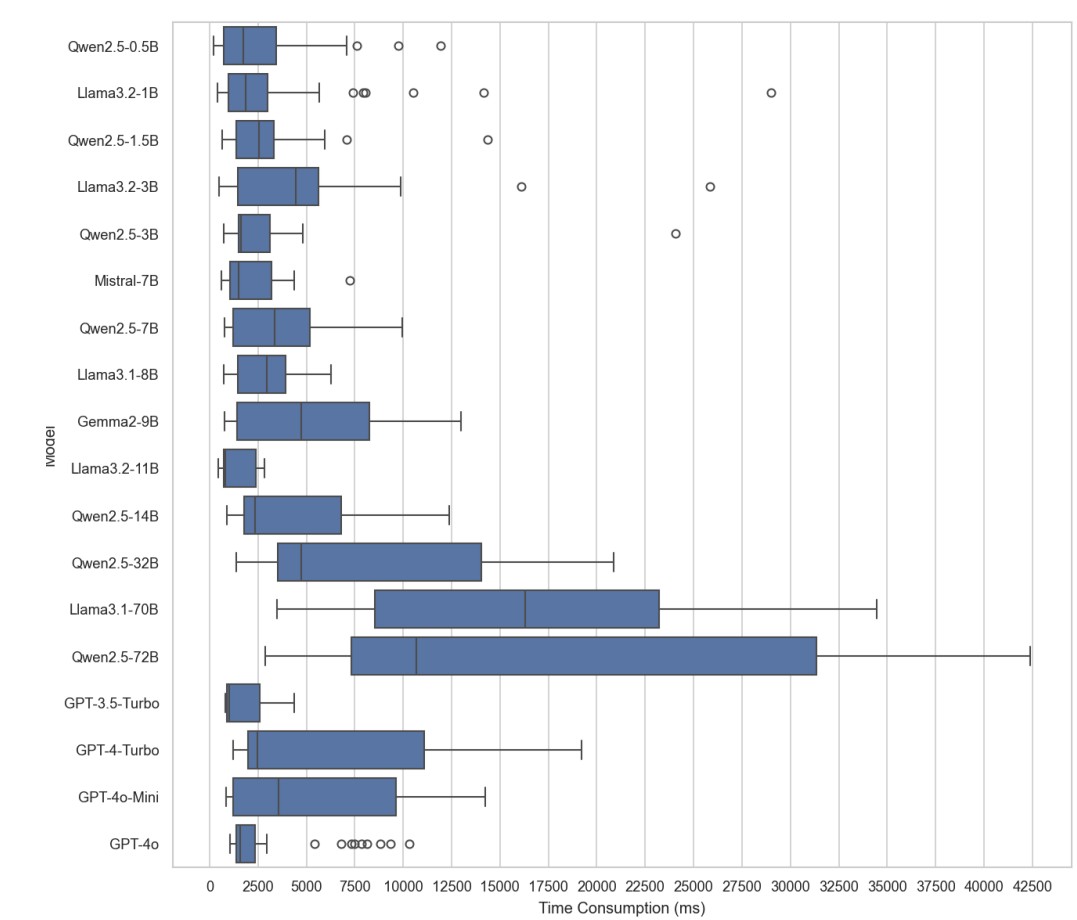

Figure 7: Box plot of time consumption for training with one data entry. This graph can be used to roughly estimate the time consumption in actual application.

## C  EVALUATION ON SUBSTITUTION SCRIPT

Table 5 shows the performance and the corresponding change in performance of *Mockingbird* when the substitution script feature is enabled. The performance is evaluated on the same classification tasks as in the performance evaluation (section 3.1) with a context length of 40. Other configurations remain the same as for the performance evaluation.

In table 5, it is noteworthy that there is an significant accuracy increase on accuracy when substitution script feature is enabled for "Poisonous Mushrooms Classification" task when the context length is 0. The accuracy is increased from 0.3720 to 0.8970. Its content is shown as code 2. Unlike the example that we discussed in appendix E, the LLM did not pay much attention to almond odor in this script. Also, this script is only tested on 1000 data entries, so it is possible for such one simple rule to classify poisonous mushrooms with a relatively high accuracy.

Their performance on regression tasks with continuous values drops significantlty as expected. Code 3 is the substitution script generated for the "Car Price Regression" task. From this code we can see that only 4 variables are used, and all these variables are either numeric or boolean; the remaining text variables are ignored because it is hard for the LLM to process in static script. However, when working in dynamic model, LLMs can benefit from these text fields such as model name of the car ("model" field) and engine name ("engine" field). This also confirms our statement that compared to the code generation solution, using LLMs as learning machines can benefit from

Table 4: Token consumption and estimated monetary cost for several LLMs with different parameters. Consumption is categorized into "invocation consumption" and "reflection consumption"; "usage" represents the count of consumed tokens; "cost" represents the estimated money cost in US dollars for the corresponding token usage.

| Models | Invocation Usage | Reflection Usage | Token Price($/M) | Total Tokens | Total Cost ($) |
|---|---|---|---|---|---|
| Gemma-2-9B | 1,772,985 | 39,393 | 0.30 | 1,812,378 | 0.54 |
| Mistral-7B | 1,867,783 | 38,019 | 0.20 | 1,905,802 | 0.38 |
| Llama-3.1-8B | 1,821,480 | 42,123 | 0.18 | 1,863,603 | 0.34 |
| Llama-3.1-70B | 1,881,546 | 30,826 | 0.88 | 1,912,372 | 1.68 |
| Llama-3.2-1B | 2,040,186 | 45,579 | 0.06 | 2,085,765 | 0.13 |
| Llama-3.2-3B | 1,569,859 | 50,968 | 0.06 | 1,620,827 | 0.10 |
| Llama-3.2-11B | 1,769,958 | 36,416 | 0.18 | 1,806,374 | 0.33 |
| Qwen-2.5-0.5B | 1,807,772 | 50,882 | 0.10 | 1,858,654 | 0.19 |
| Qwen-2.5-1.5B | 1,951,434 | 45,660 | 0.10 | 1,997,094 | 0.20 |
| Qwen-2.5-3B | 1,824,880 | 27,296 | 0.10 | 1,852,176 | 0.19 |
| Qwen-2.5-7B | 1,856,441 | 30,455 | 0.30 | 1,886,896 | 0.57 |
| Qwen-2.5-14B | 1,838,653 | 27,479 | 0.30 | 1,866,132 | 0.56 |
| Qwen-2.5-32B | 1,863,262 | 29,306 | 0.80 | 1,892,568 | 1.51 |
| Qwen-2.5-72B | 1,921,516 | 34,331 | 1.20 | 1,955,847 | 2.35 |
| GPT-3.5-Turbo | 6,744,766$^\uparrow$  43,237$^\downarrow$ | 57,542$^\uparrow$  1,527$^\downarrow$ | 0.50$^\uparrow$  1.50$^\downarrow$ | 6,802,308$^\uparrow$  44,764$^\downarrow$ | 3.47 |
| GPT-4-Turbo | 9,524,486$^\uparrow$  42,185$^\downarrow$ | 74,491$^\uparrow$  4,898$^\downarrow$ | 10.00$^\uparrow$  30.00$^\downarrow$ | 9,598,977$^\uparrow$  47,083$^\downarrow$ | 97.40 |
| GPT-4o | 8,283,508$^\uparrow$  53,680$^\downarrow$ | 51,037$^\uparrow$  2,898$^\downarrow$ | 2.50$^\uparrow$  10.00$^\downarrow$ | 8,334,545$^\uparrow$  56,578$^\downarrow$ | 21.40 |
| GPT-4o-Mini | 12,648,541$^\uparrow$  96,856$^\downarrow$ | 133,514$^\uparrow$  6,777$^\downarrow$ | 0.15$^\uparrow$  0.60$^\downarrow$ | 12,782,055$^\uparrow$  103,633$^\downarrow$ | 1.98 |

$^\uparrow$ Price or consumption for input tokens.
$^\downarrow$ Price or consumption for output tokens.

Table 5: Accuracy evaluation on tasks when the substitution script feature is enabled. Columns are configurations with different context length, where context length 0 means zero-shot. For classification tasks, "+/-" represents the change of accuracies; for regressions tasks, "+/-" represents the inverse of change in RMSE (root mean square error) , so that negative values indicate that the performance is worsen; these changes are calculated in comparison to those in table 1.

| Dataset | 0 | +/- | 20 | +/- | 40 | +/- | 60 | +/- | 80 | +/- |
|---|---|---|---|---|---|---|---|---|---|---|
| Titanic | 0.7317 | -0.0562 | 0.6647 | -0.1240 | 0.7332 | -0.0411 | 0.7268 | -0.0650 | 0.7706 | -0.0160 |
| Mushrooms | 0.8970 | +0.5250 | 0.1020 | -0.4673 | 0.1010 | -0.6292 | 0.8957 | -0.1011 | 0.8989 | +0.0630 |
| Horses | 0.4600 | -0.0850 | 0.3673 | -0.1858 | 0.4645 | -0.1191 | 0.4638 | -0.0894 | 0.4532 | -0.0555 |
| Insurances | 0.7160 | +0.1280 | 0.8602 | +0.1872 | 0.8031 | +0.0771 | 0.7297 | +0.0308 | 0.6913 | +0.0120 |
| Car Prices | 109,437 | -87,212 | 112623 | -90100 | 113904 | -84989 | 114969 | -90435 | 116365 | -93571 |
| Mohs Hardness | 1.8553 | -0.5750 | 1.9683 | -0.8087 | 6.9817 | -5.8503 | -$^*$ | -$^*$ | -$^*$ | -$^*$ |

$^*$ This dataset only contains 57 data entries.

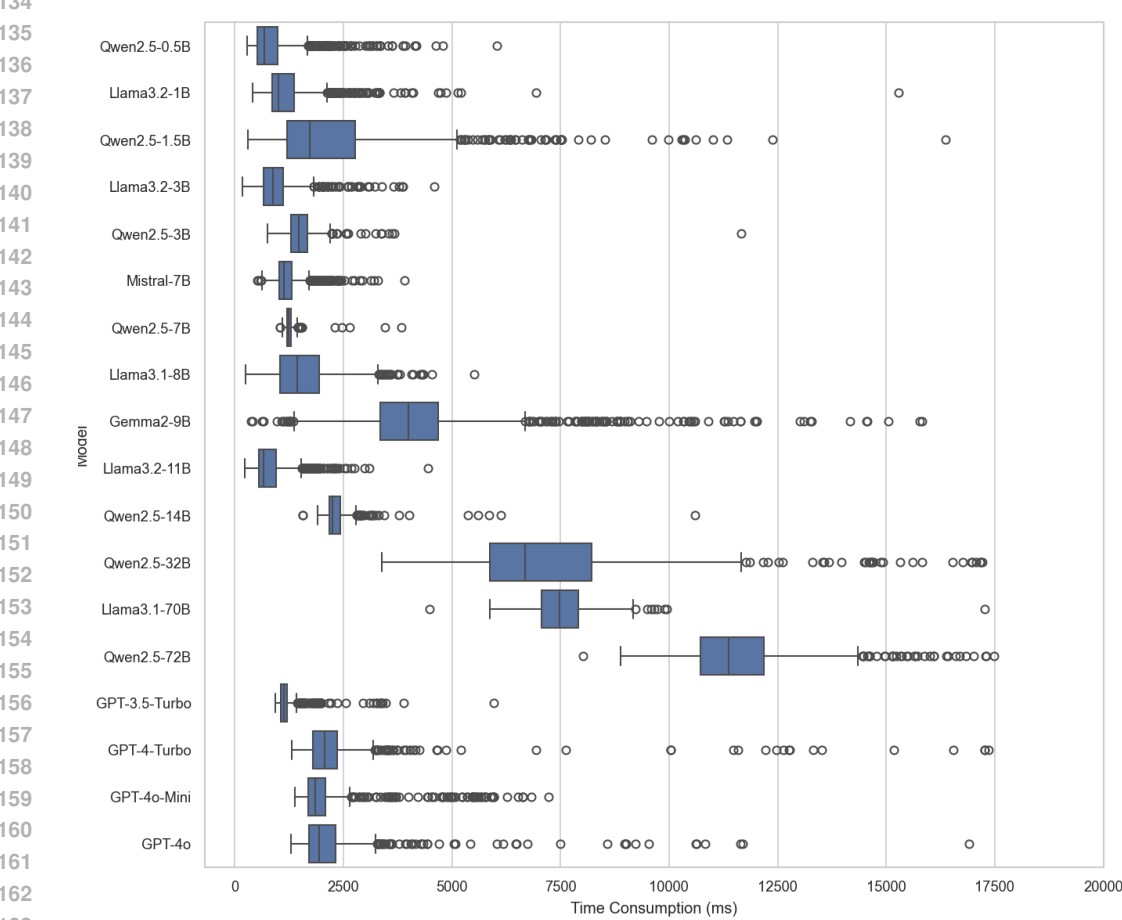

Figure 8: Box plot of time consumption for evaluating with one data entry. This graph can be used to roughly estimate the time consumption in actual application.

their semantic understanding ability and therefore benefit from those data fields that are difficult to perform conventional numerical computations on.

## D    EVALUATION WITH RETRIEVAL-AUGMENTED GENERATION

In this section, we evaluate the performance of *Mockingbird* with retrieval-augmented generation (RAG), which can intuitively demonstrate its advantages in the field of machine learning. The performance with RAG is shown in table 7, compared with the performance without RAG in table 3.

To simulate the RAG process, we fetch several texts from the "Titanic" webpages on *Wikipedia*[13] and *Encyclopedia Titanica*[14]. And we categorize them into 3 category of levels, according to how much we subjectively feel they may contribute to the performance:

**Level 1**    Strong hints. Texts in this category directly reveals the statistic data of the survivals such as the table 6.

**Level 2**    The list of survivor names.

From the performance comparison shown in table 7, we have observed an unexpected finding: not all models used the survivor list provided in the level 2 RAG materials. Text 2 is the reasoning

---

[13]URL: https://en.wikipedia.org/wiki/Titanic, data captured on Nov. 25[th], 2024.
[14]URL: https://www.encyclopedia-titanica.org/titanic-survivors/, data was captured on Nov. 25[th], 2024.

Table 6: The table of survival statistic data which is injected into the context as level 1 RAG material. This table is injected into the context in the format of Markdown table.

| Sex/Age | Class/Crew | Number Aboard | Number Saved | Number Lost | Percentage Saved | Percentage Lost |
|---------|-----------|---------------|--------------|-------------|------------------|-----------------|
| Children | First Class | 6 | 5 | 1 | 83% | 17% |
| | Second Class | 24 | 24 | 0 | 100% | 100% |
| | Third Class | 79 | 27 | 52 | 34% | 66% |
| Women | First Class | 144 | 140 | 4 | 97% | 3% |
| | Second Class | 93 | 80 | 13 | 86% | 14% |
| | Third Class | 165 | 76 | 89 | 46% | 54% |
| | Crew | 23 | 20 | 3 | 87% | 13% |
| Men | First Class | 175 | 57 | 118 | 33% | 67% |
| | Second Class | 168 | 14 | 154 | 8% | 92% |
| | Third Class | 462 | 75 | 387 | 16% | 84% |
| | Crew | 885 | 192 | 693 | 22% | 78% |
| Total | | 2224 | 710 | 1514 | 32% | 68% |

Table 7: Performance of different LLMs with different levels of RAG materials. The column "Context" represents the accuracy under a given context length; "Context 0" represents the accuracy when the context length is 0, which is a zero-shot configuration. The column "+/-" represents the change of accuracy compared to the accuracy without RAG shown in table 3. "Time (ms) +/-" represents the change of average time consumption in seconds; "Token +/-" represents the change of token consumption; "Cost ($) +/-" represents the change of estimated money cost for token consumption in US dollars.

| Level | Model | Context 0 | +/- | Context 40 | +/- | Time (s)[†] +/- | Token[†] +/- | Cost ($)[†‡] +/- |
|-------|-------|-----------|-----|------------|-----|-----------------|--------------|-------------------|
| 1 | GPT-4o | 0.7946 | +0.0113 | 0.7920 | -0.0188 | +0.1124 | +366,257 | +0.9156 |
| | GPT-4o-Mini | 0.7923 | +0.0325 | 0.7990 | +0.0000 | -0.5690 | +415,418 | +0.0623 |
| | Qwen-2.5-72b | 0.8013 | +0.0180 | 0.7920 | +0.0118 | +11.3635 | +426,406 | +0.5117 |
| | Qwen-2.5-32b | 0.8035 | +0.0291 | 0.8037 | +0.0035 | +2.8397 | +421,312 | +0.3370 |
| | Qwen-2.5-14b | 0.7991 | +0.0225 | 0.8014 | -0.0117 | -0.0773 | +425,041 | +0.1275 |
| 2 | GPT-4o | 0.9887 | +0.2008 | 0.8719 | +0.0611 | +0.2749 | +3,225,212 | +8.0630 |
| | GPT-4o-Mini | 0.7867 | +0.0269 | 0.7837 | -0.0153 | -0.2884 | +3,276,431 | +0.4915 |
| | Qwen-2.5-72b | 0.7766 | -0.0067 | 0.8049 | +0.0247 | -0.3253 | +21,466 | +0.0258 |
| | Qwen-2.5-32b | 0.7856 | +0.0112 | 0.8190 | +0.0188 | -0.1390 | +28,723 | +0.0230 |
| | Qwen-2.5-14b | 0.7474 | -0.0292 | 0.7802 | -0.0329 | -0.1792 | -26,891 | +0.0081 |

[†] This column is intended to give users a general idea of the cost of injecting RAG materials; therefore, so values in this column are for comparison when the context length is 0.
[‡] The change in token consumption is largely caused by the injection of RAG material into the context, therefore the change in estimated price is calculated according to the price of the input tokens.

```
Code 2:    Substitution Script for Mushrooms Classification

using MongoDB.Bson;
public static BsonDocument MockFunction(BsonDocument Arguments)
{
    // Extract necessary properties from the arguments
    string capShape = Arguments["capShape"].AsString;
    string capSurface = Arguments["capSurface"].AsString;
    string capColor = Arguments["capColor"].AsString;
    string bruises = Arguments["bruises"].AsString;
    string odor = Arguments["odor"].AsString;
    // Additional properties can be accessed similarly...
    // For simplicity, we check the odor to decide if it's
        poisonous or edible
    // Common domain knowledge: certain odors like 'foul' or
        'fishy' might indicate poisonous
    string results = (odor.ToLower() == "foul" || odor.ToLower()
        == "fishy") ? "Poisonous" : "Edible";
    string remarks = "Based on the odor of the mushroom which
        indicates whether it is likely to be poisonous or edible.";
    return new BsonDocument
    {
        { "Remarks", remarks },
        { "Results", results },
        { "IsReadyToCompile", true }
        // We assume the model is simplistic and script is ready
            to compile
    };
}
```

content generated by *GPT-4o* with RAG; the red marked sentence clearly indicates that survivor list is referenced in the reasoning. However, such signs cannot be found in the reasoning content generated by *GPT-4o-Mini* with RAG, as shown in text 3. This situation indicates that an explicit prompt to use RAG materials is needed for these models with small parameter sizes. We also found that the level 1 RAG materials bring the accuracy of the LLMs to around 0.8000%, regardless of their parameter sizes. It is usually believed that LLMs with more parameters can carry more intrinsic knowledge, it seems that these RAG materials fills the gap in the accuracy and richness of the intrinsic knowledge of these LLMs with fewer parameters.

Another finding is that the cost of RAG is generally acceptable. When using commercial LLM inference services such as the *GPT* series the average time consumption does not change significantly. Although the change in token consumption is huge, the change is largely caused by input tokens of RAG material with a relatively low price, so the change in cost is still acceptable.

---

Text 2:    Example of Remarks Generated by GPT-4o with RAG

The passenger does not appear on the list of known survivors provided. Additionally, Mr. Braund was a young male in third class with a low ticket price and no recorded cabin, which historically had lower survival rates on the Titanic.

- - - - - - - - - - - - - - - - - - - - - - - - - - - - - - - - - - - - - - - - - - - - - - - - -

Log item 6746b2572950a558068a689b, in *"Titanic-Level3-GPT-4o-Context_0-98.8777%.json"*

---

**Code 3:** Substitution Script for Car Price Regression

```csharp
using MongoDB.Bson;
public static BsonDocument MockFunction(BsonDocument Arguments)
{
    int basePrice = 20000;  // Base price for estimation
    int depreciation = (2023 - Arguments["modelYear"].AsInt32) *
        1000;
    string mileage = Arguments["milage"].AsString;
    int mileageValue;
    int.TryParse(mileage.Replace("km", "").Trim(), out
        mileageValue);
    int mileagePenalty = (mileageValue / 10000) * 500;
    int accidentPenalty = Arguments["accident"].AsString.ToLower()
        == "yes" ? 3000 : 0;
    int cleanTitleBonus = Arguments["cleanTitle"].AsBoolean ? 2000
        : 0;
    int estimatedPrice = basePrice - depreciation - mileagePenalty
        - accidentPenalty + cleanTitleBonus;
    string remarks = "Estimated price based on model year,
        mileage, accident history, and clean title.";
    var result = new BsonDocument
    {
        { "Remarks", remarks },
        { "Results", estimatedPrice },
        { "IsReadyToCompile", true }
    };
    return result;
}
```

---

**Text 3:** Example of Remarks Generated by GPT-4o-Mini with RAG

Mrs. Johnson, despite being in third class and having no siblings on board, had two children with her. Women and children were prioritized during evacuation, which increases her chance of survival compared to male passengers of the same class.

- - - - - - - - - - - - - - - - - - - - - - - - - - - - - - - - - - - - - - - - - - - - - - -

Log item 6746bdf8881a81ae902421d7, in "*Titanic-Level3-GPT-4o-Mini-Context_0-78.6756%.json*"

---

We anticipate that there may be some controversy about the significance of this experiment. However, the fact is that, in many real-world scenarios, it is possible to access such non-structural information that can contribute to the task or even reveal the answer. For one example, in the classic task of credit card issuing task, where models should predict that whether the applicant will default on credit in the future, this non-structural information such as personal activity history, public information on the social network applications, could indicate the financial status of the applicant and thus improve the credibility of the prediction results. As another example, in the task of short-term stock price prediction task, news about the company which issuing the stock in fact can indicate the change of the stock price in the short term. By installing web browsing plugins, LLMs can benefit from this real-time non-structural information to have a chance to make a relatively more solid prediction.

## E    DISCUSSION ON THE POSSIBLE LIMITATIONS IMPOSED BY THE INTRINSIC KNOWLEDGE

The "Poisonous Mushrooms Classification" task shown in table 1 can be a good example to illustrate the potential limitations that may be imposed by the intrinsic knowledge of LLMs. In this task, the

scores with insufficient training (when context lengths are 0 and 20) are very low, and when the context context is 0 (zero-shot), the score is 0.3720 which is even below the accuracy of random guessing (0.5). When there is no training procedure, the LLM relies only on its intrinsic knowledge to perform the reasoning, however, sometimes the intrinsic knowledge that the LLM has gained from pre-training data is not "true" for this task.

For example, by comparing the "remarks" field (reasoning) in the operation logs for context length 0 (when accuracy is 0.3720) and context length 80 (when accuracy is 0.8359), we found that the LLM's understanding of the correlation between mushroom almond odor and toxicity changed significantly after training. In the reasoning process without training, the LLM believed that almond odor was a sign of poisonous in text 4.

---

**Text 4:   Remarks for an Invocation When Context Length is 0**

The mushroom has an almond odor, which is **often associated with poisonous species**. Moreover, characteristics such as its white spore print and its environment lead to the conclusion that it is not safe for consumption.

- - - - - - - - - - - - - - - - - - - - - - - - - - - - - - - - - - - - - - - - - - - - - - - - -

Log item 66fc730b8b1eb660f79c42d8, in
"*Mushrooms-Context_0-Training_0-37.20%.json*".

---

However, after sufficient training, it believed that almond odor was more often associated with edible species in text 5. Despite of this difference in the understanding of mushroom odor, we have also observed that: after training, the LLM began to consider various factors more comprehensively (such as odor, cap, spore sprint in the example) instead of relying too much on the single factor of odor.

---

**Text 5:   Remarks for an Invocation When Context Length is 80**

The almond-like odor, yellow cap, and spore print characteristics are **commonly associated with edible mushrooms**. These features suggest the mushroom is likely safe for consumption.

- - - - - - - - - - - - - - - - - - - - - - - - - - - - - - - - - - - - - - - - - - - - - - - - -

Log item 66fc944a4e779f389770f298, in
"*Mushrooms-Context_60-Training_60-99.7872%.json*".

---

# F   DISCUSSION ON APPLYING MOCKINGBIRD TO SPECIFIC DOMAINS

In the "Horse Colic" task in table 1, the reflection mechanism is observed to be almost ineffective.

Based on the findings that current LLMs cannot discover the errors within their reasoning process in the absence of external feedback (Huang et al., 2024a), we speculate that the reason it cannot learn on this task is that it is not really "learning" on this task. The only external feedback in the training stage is the ground truth from the dataset; however, the ground truth can only indicate the existence of errors within the reasoning flows; what exactly these errors are requires LLMs to figure out based on their knowledge and reasoning. If their intrinsic knowledge does not cover this specific domain, then chances are that they will not be able to reason out the errors within their previous reasoning process and consequently will not improve throughout the training stage. For example, text 6 and text 7 are reflection notes generated by *GPT-4o* for the "Horse Health Prediction" task.

Then, it made an error by outputting wrong answer "Lived" instead of the correct answer "Died". The corresponding reflection hint is as text 7.

There are two major problems in these two reflection notes:

1) Lack of concrete actionable tips. The content marked in red in these reflection notes are too abstract to be tips. They are more like directional requirements about "what to do", but lack the most important content of "how to do".

---

Text 6: Reflection Note for Horse Health Dataset - Sample 1

I previously given the wrong result Died, but the correct answer is Lived.

Mistake Analysis:

1.Overemphasis on Immediate Critical Signs: I focused heavily on the immediate critical signs like severe abdominal distension and absent peristalsis, interpreting them as insurmountable without considering the potential for successful intervention through surgery.

2. Insufficient Weight on Post-Surgical Recovery: The fact that surgery was performed was underplayed in the outcome analysis. Surgical intervention can often have a significantly positive effect, suggesting a greater potential for recovery than what was initially considered.

3. Neglecting Positive Indicators: There were positive signs like normal extremity temperature and less than 3 seconds capillary refill time which were overlooked in weighing the overall prognosis. These signs are indicators of stability post-surgery.

Notes for Future Reference:

1. Balance Critical and Positive Indicators: Evaluate both critical and positive indicators equally to form a more balanced prognosis. Recognize the role of surgical intervention and how it can improve outcomes even in severe cases.

2. Consider Surgical Success: If surgery has been performed, consider the potential success of the surgical intervention and recovery rate statistics which may lean towards a positive outcome in many cases.

3. Utilize All Relevant Indicators: Include all available clinical signs in the assessment, giving weight to those that may indicate stability or improvement, even when severe issues are present.

4. Factor in Recent Interventions: Always consider the timeline and effects of recent interventions like surgery more carefully, as these can significantly influence outcomes in favor of recovery.

By integrating these notes into future assessments, I can improve accuracy in predicting health outcomes, recognizing that intervention and recovery are possible even amidst severe symptoms.

- - - - - - - - - - - - - - - - - - - - - - - - - - - - - - - - - - - - - - - - - - - -

Log item 66fcce19646903bc4fba13ed, in *"Horses-Context_80-Training_80-50.8696%.json"*.

---

2) Lack of quantitative facts. A positive example is "positive signs ... less than 3 seconds capillary refill time" in sample 1. We have verified this as a fact, the normal capillary refill time of horses is about $1 \sim 2$ seconds[15]. These quantitative facts can help LLMs make deterministic judgments.

There are currently a number of methods that can alleviate this problem to some extent.

The first type of method is to provide LLMs with external domain-specific knowledge. This can be achieved by conventional retrieval-augmented generation with documents in the specific domain, or by wrapping connectors to web search engines or databases containing domain-specific knowledge as tools for LLMs, thus allowing LLMs to search related information at inference time.

The second type of method is to introduce human feedback into the reasoning process of LLMs. During the training stage, human experts can review the reasoning content and the reflection notes, and directly edit them if these experts deem it necessary. With this approach, domain-specific knowledge is proactively injected into the context, which can guarantee that LLMs pay attention to some extent.

The first type of passive methods is more suitable for development teams that lack domain experts. The second type of active supervision-based methods can speed up the learning process to some extent in the early training stage, and provide more control over the behavior of the mock functions learned from training.

---

[15]Horse Side Vet Guide - Database Record Viewer - Capillary Refill Time (CRT) Prolonged - URL: https://horsesidevetguide.com/drv/Observation/80/capillary-refill-time-crt-prolonged/

---

Text 7:   Reflection Note for Horse Health Dataset - Sample 2

I previously given the wrong result Lived, but the correct answer is Died.

Mistake Analysis:

1. Misinterpretation of Indicators: I placed too much emphasis on stable indicators like normal temperature of extremities and peripheral pulse while not fully realizing the severity suggested by other signs, such as the elevated pulse, respiratory rate, and underlying conditions indicated by surgical lesions.

2. Underestimation of Surgical Implications: The assumption that surgical intervention would result in recovery was overly optimistic. I failed to account for the possibility that despite surgery, the horse could succumb to its condition if the underlying problem or the surgical aftermath was too severe.

3. Overemphasis on Positive Signs: I focused largely on the normal physiological indicators and underappreciated the significance of surgical lesions and critical symptoms that could contribute to a poor outcome despite normal readings elsewhere.

Notes for Future Reference:

1. Evaluate Severity Holistically: All indicators, especially those pointing to potential systemic failure or severe distress, should be carefully weighed against positive signs. Consider how severe symptoms might overshadow stable vital signs when predicting outcomes.

2. Understand Surgery Limitations: Recognize that while surgery can address certain conditions, it may not guarantee recovery if severe systemic or non-reversible complications are present. Assess the potential outcomes of surgery more critically.

3. Consider Comprehensive Context: Factor in the entire clinical picture, including post-surgical risks and the severity of existing conditions, even when some signs appear normal or stable.

4. Examine Critical Vital Signs Thoroughly: Higher pulse and respiratory rates, severe abdominal distension, and other serious signs should prompt careful consideration of a possibly poor prognosis even when other parameters seem normal.

By integrating these notes, I can better balance my assessment of both positive and negative indicators, leading to more accurate health outcome predictions in complex veterinary cases.

- - - - - - - - - - - - - - - - - - - - - - - - - - - - - - - - - - - - - - - - - - - - -

Log item 66fcce23646903bc4fba13ee, in "*Horses-Context_80-Training_80-50.8696%.json*".

## G    GUIDELINES FOR DEPLOYMENT IN RESOURCE CONSTRAINED ENVIRONMENTS

Based on the results of our experiments, here are our recommendations and tips for using *Mockingbird* in resource-constrained environments:

- Prioritize models with fewer parameters and higher formal correctness ratio. Some models with small parameter sizes have a lower formal correctness ratio, which will lead to re-generating responses and eventually a higher token and time consumption. If such models are not in the options, consider adding at least a few example invocations to teach LLMs the schema of inputs and outputs; according to our experiments above (table 3), training process can usually significantly improve the formal correctness ratio and thus improve the system stability.

- Prioritize proactive human revision on remarks and reflection notes over providing more but less relevant documents as RAG materials. As shown in previous experiments on RAG (table 7), smaller models usually have a difficulty making full use of RAG materials, so proactive human revision may be more efficient in improving performance.

- If it is a necessary to use commercial LLM inference services, favor these providers that have lower prices for input tokens, since most of the token consumption for *Mockingbird* is input tokens (uploading arguments and RAG materials); unless the task has an obvious feature that the output tokens will be significantly more than the input tokens.

- Use models with large parameter sizes as reflectors. *Mockingbird* allows users to configure different models as "executors" (performing the reasoning and generating results), "reflectors"

(reflecting on previous errors and generating reflection notes) and "generators" (generating substitution scripts). If resources are limited, consider using models with a large parameter size to make the training process more effective, if conditions permit.

- Periodically perform the training process. The training process can be performed at any time, rather than only prior to deployment. It is recommended to collect invocation data at run-time, and periodically (daily, weekly, monthly, etc.) to update the behavior of mock functions to keep matching the situation.

