# OpenReview forum: "Mockingbird: Platform for Adapting LLMs to General Machine Learning Tasks"
_ICLR.cc/2025/Conference — Submitted to ICLR 2025_

### Official Review · Reviewer_hhzP · 2024-10-31

**Soundness:** 2
**Presentation:** 2
**Contribution:** 2
**Rating:** 5
**Confidence:** 4

**Summary:**

This research paper proposes a new paradigm called Mockingbird, which leverages large language models (LLMs) as "mock functions" to adapt them to general machine learning tasks. Mockingbird allows users to define mock functions, which are functions without code bodies but only with method signatures and documentation. At runtime, Mockingbird instructs the LLM to "role-play" these mock functions, enabling LLMs to perform tasks beyond traditional chatbot interactions. The paper discusses how this paradigm offers advantages over conventional machine learning methods, including handling zero-shot scenarios, adapting to fluctuating data fields, and utilizing external sources like the internet. The authors present an open-source platform implementing Mockingbird and evaluate its performance on various datasets from Kaggle, demonstrating its competitive capabilities.

**Strengths:**

This is a framework paper with good intuition.
1. The authors effectively use figures to illustrate key concepts, such as the Mockingbird workflow and memory management features. It demonstrates a good system design.
2. The paper demonstrates a good understanding of related work in LLMs, in-context learning, and code generation, with a comprehensive list of references.
3. The authors provide sufficient detail about the implementation, including system prompt construction and JSON schema validation.
4. The branch control feature of mock memories is fascinating. This LLM agent's memory update feature could also be applied to other memory-based agent frameworks to enhance consistency and coherence.

**Weaknesses:**

This paper has several weaknesses regarding the description of its methodologies and the scope of experiments conducted.
1. The author did not explain several key components regarding the memory systems and final execution.
    - The description of Section 2.4 is hard to follow. From the first paragraph, I can read about the intuition behind compressing and replacing memory, but how to conduct that memory compression is undefined. It would be helpful to provide a detailed description of your implemented memory compression algorithm or to include pseudocode for their approach, as the current description sounds more like a survey and compilation of different methods without a specific implementation.
    - If possible, the authors could create another figure to show how a substitution script is generated. I am extremely interested in what type of substitution script looks like. Is it a rule-based decision system with if-else statements? Experiments of the drop in accuracy after the translation are necessary, even if the authors believe this step is optional.
    - If all the problems are binary classification and regression containing a single answer, why is JSON schema output necessary? I am not saying there are specific drawbacks, but introducing JSON schema for this simple scenario is confusing. Why don’t we just prompt the LLM as a multi-choice QA?
2. Some of the claims in Section 3.2 Discussion do not convince me. The limited scope of the experiments conducted also raises some concerns.
    - All experiments are conducted within small Kaggle datasets. Several experiments showed poor performance against humans. We should either improve the performance to a reasonable point or include a more in-depth analysis of why MockingBird fails.
    - No comparisons of different base models are included. The authors recommend that the user fine-tune self-host LLMs, but it seems they were not responsible for this claim. If all the tasks are binary classification or regression, it would not be difficult to finetune an LLaMA-3 8B model to generate reasonable feedback. For example, experiments for closed-source models (i.e., GPT-4o-mini, Claude-3.5-Sonnet) and open-source models (i.e., Qwen-1.5,  LLaMA-3, Deepseek, and Mistral) would be helpful. Some models on Together AI do support JSON mode [1].
    - Longer context is not all you need. This analysis should not be drawn from your experiment results. The discovery, however, seems to be interesting, as a conclusion might be drawn from the intrinsic properties of these Kaggle tasks. One or two concrete examples from each experiment should be added to assist the analysis. It should be the properties of the tasks that lead to this phenomenon, not the phenomenon itself (i.e., longer context degrades performance in some tasks) that lead to this phenomenon. For example, illustrations of how LLM missed correct information among longer in-context examples may help.
    - Why Cannot LLMs Be 'a Good Veterinarian'? The reasoning is somewhat unclear, and the ideas jump between.
      - The paper suggests that the higher number of data fields in the "Horses" task contributes to the difficulty. Explain why this might be the case. Could it be related to the increased complexity of relationships between variables, the potential for more missing data, or the increased difficulty in identifying relevant features?  You could explore the relationship between the number of data fields and task difficulty and provide concrete examples of how the "guessing trap" manifests in this task. This will be clearer if a trend of increased difficulty with more data fields is observed (i.e., a plot showing the decrease in performance).
      - The explanation of the "guessing trap" is too abstract. Use concrete examples from the "Horses" task to illustrate how the LLM might be making incorrect assumptions about its errors and how those incorrect assumptions lead to a lack of improvement. For example, you might outline the steps of their reasoning, supported by specific observations from your experiments on the "Horses" task.
3. A cost analysis is necessary for each task run. It seems to be extremely expensive to run one experiment with Mockingbird. Factors such as GPU hours, number of API calls (if using a commercial LLM service), and estimated financial cost will be helpful.
4. More ablation studies regarding each component of MockingBird are helpful. For example, could using serializers for both input and output boost performance? How and what information should be compressed within the memory compression steps? Or should we use a monkey patch substitution script directly generated by LLM in the training loop? Making two tables with suboptimal results is never enough for an ICLR-level publication.

[1] Together AI. JSON Mode. https://docs.together.ai/docs/json-mode. Accessed: 2024-10-30.

**Questions:**

In Section 2.4 Memory Replacing and Compression, should I interpret $mn^2 + 0.5mn$ as $\frac{1}{2}mn(n+1)$? I did not get why we have a $0.5mn$ term.

The ideology of treating LLM call as a function is not new [1].

From the supplementary material, is the original code for execution not included?

More works adapt LLMs to machine-learning tasks [2, 3, 4, 5], though not all are relevant to Kaggle competitions.

[1] Lin, C., Han, Z., Zhang, C., Yang, Y., Yang, F., Chen, C., \& Qiu, L. (2024, July). Parrot: Efficient serving of LLM-based applications with semantic variables. Presented at the 18th USENIX Symposium on Operating Systems Design and Implementation (OSDI 24), Santa Clara, CA. USENIX Association.

[2] Guo, S., Deng, C., Wen, Y., Chen, H., Chang, Y., \& Wang, J. DS-Agent: Automated Data Science by Empowering Large Language Models with Case-Based Reasoning. In Forty-first International Conference on Machine Learning.

[3] Liu, Y., Tang, X., Cai, Z., Lu, J., Zhang, Y., Shao, Y., ... \& Gerstein, M. (2023). ML-Bench: Large Language Models Leverage Open-source Libraries for Machine Learning Tasks. arXiv e-prints, arXiv-2311.

[4] Huang, Q., Vora, J., Liang, P., \& Leskovec, J. (2023). Benchmarking Large Language Models as AI Research Agents. In NeurIPS 2023 Foundation Models for Decision Making Workshop.

[5] Hollmann, N., Müller, S., \& Hutter, F. (2024). Large Language Models for Automated Data Science: Introducing CAFFE for Context-Aware Automated Feature Engineering. Advances in Neural Information Processing Systems, 36.

---

> ### Author Response · Authors · 2024-11-18
>
> Thank you for reviewing our work!
>
> >"In Section 2.4 Memory Replacing and Compression, should I interpret mn^2+0.5mn as (1/2)mn(n+1)? I did not get why we have a 0.5mn term."
> >
> >"1. A cost analysis is necessary for each task run. It seems to be extremely expensive to run one experiment with Mockingbird. Factors such as GPU hours, number of API calls (if using a commercial LLM service), and estimated financial cost will be helpful."
>
> Sorry we have a typo here. The cost of carrying samples should be 0.5mn^2 +0.5mn, which is mathematically equivalent to (1/2)mn(n+1). We want to more intuitively show that there is a nonlinear component (square) in the cost. Still, this is just a vague estimate of expenses; actual expenses will be higher than this.
> As commented by you and reviewer 4oeR, we in deed should include a cost analysis to help users estimate the cost of money and time.
>
> Currently we are: (a) performing cost analysis on the operation logs of existing experiments; (b) developing a method for Mock Trainer to estimate the money and time cost before it starts; (c) adding summary of cost information into operation logs.
> We will keep reporting our progress regarding this issue.
>
> > "From the supplementary material, is the original code for execution not included?"
>
> Yes, it is true; we released our source code in the latest supplementary material. Mockingbird relies on an early version of our unpublished work, Robotless Platform. Still, we failed to find a proper way to separate them apart, so in the latest version of supplementary material, we uploaded the source code of both Robotless and Mockingbird.
>
> The source code and a reproduction guide are now included in the latest version of supplementary material.
> We sincerely do not recommend to use other features of this under-development version of Robotless, and please note that this version of Robotless does not imply its final performance nor implementation.
> We will provide more help regarding Robotless once we are ready.
>
> >"No comparisons of different base models are included."
> >
> >"For example, experiments for closed-source models (i.e., GPT-4o-mini, Claude-3.5-Sonnet) and open-source models (i.e., Qwen-1.5, LLaMA-3, Deepseek, and Mistral) would be helpful. Some models on Together AI do support JSON mode."
>
> It's a good suggestion. Since all reviewers mentioned the experiment of scalability, we believe that our potential users may also want to know its scalability. Sorry, we overlooked this before.
> Currently we are conducting experiment of scalability on TogetherAI, and we found it indeed convenient to use, thank you for your recommendation.
> This experiment is a little time-consuming, and we will immediately report our result once it is done.

---

> ### Author Response · Authors · 2024-11-18
>
> > "If possible, the authors could create another figure to show how a substitution script is generated. I am extremely interested in what type of substitution script looks like. Is it a rule-based decision system with if-else statements?"
>
> It is our mistake that we didn't give sufficient explanation on substitution script, and we will add a figure and extra text to fully explain it in the revised paper.
> Substitution script is the code that LLMs generate to implement a mock function, sometimes it is written in if-else statements, and sometimes it is rule-based - this depends on the type of output.
> This is one version of the substitution script for Iris dataset:
>
> ```csharp
> using MongoDB.Bson;
>
> public static BsonDocument MockFunction(BsonDocument Arguments)
> {
>     double sepalLength = Arguments["sepalLength"].ToDouble();
>     double sepalWidth = Arguments["sepalWidth"].ToDouble();
>     double petalLength = Arguments["petalLength"].ToDouble();
>     double petalWidth = Arguments["petalWidth"].ToDouble();
>
>     string species = "Unknown";
>     string remarks = "";
>
>     if (petalLength < 2.0)
>     {
>         species = "Setosa";
>         remarks = "Petal length is less than 2.0, indicating Setosa.";
>     }
>     else if (petalLength >= 2.0 && petalLength < 5.0)
>     {
>         if (petalWidth < 1.5)
>         {
>             species = "Versicolor";
>             remarks = "Petal length and width are within the range for Versicolor.";
>         }
>         else
>         {
>             species = "Virginica";
>             remarks = "Petal width is greater than or equal to 1.5, indicating Virginica.";
>         }
>     }
>     else if (petalLength >= 5.0)
>     {
>         species = "Virginica";
>         remarks = "Petal length is greater than or equal to 5.0, indicating Virginica.";
>     }
>     else
>     {
>         remarks = "Unable to determine species from given dimensions.";
>     }
>
>     return new BsonDocument
>     {
>         { "Remarks", remarks },
>         { "Results", species },
>         { "IsReadyToCompile", true }
>     };
> }
>
> return MockFunction(Arguments);
> ```
> And this is one version of the substituion script for Titanic dataset:
> ```csharp
> using MongoDB.Bson;
>
> public static BsonDocument MockFunction(BsonDocument Arguments)
> {
>     // Extracting data from the Arguments BsonDocument
>     int passengerClass = Arguments["passengerClass"].AsInt32;
>     string sex = Arguments["sex"].AsString;
>     int age = Arguments["age"].AsInt32;
>     int siblingsOnBoard = Arguments["siblingsOnBoard"].AsInt32;
>     int parchOnBoard = Arguments["parchOnBoard"].AsInt32;
>     double ticketPrice = Arguments["ticketPrice"].AsDouble;
>
>     // Initialize the result fields
>     double survivalChance = 0.0;
>     string remarks = "";
>
>     // Simplistic logic to determine survival chance
>     // This is based on well-known historical data patterns from the Titanic
>     if (sex.ToLower() == "female")
>     {
>         survivalChance += 0.5; // Females generally had a higher survival rate
>         remarks += "Gender favorably affects survival chances. ";
>     }
>
>     if (passengerClass == 1)
>     {
>         survivalChance += 0.3; // First-class passengers had better survival rates
>         remarks += "First class improves survival chances. ";
>     }
>
>     if (age < 12)
>     {
>         survivalChance += 0.1; // Children had a higher chance of survival
>         remarks += "Being a child increases survival odds. ";
>     }
>
>     if (siblingsOnBoard + parchOnBoard >= 3)
>     {
>         survivalChance -= 0.1; // Large families had lower survival rates
>         remarks += "Being in a large group negatively affects survival chances. ";
>     }
>
>     // Make sure the survival chance is within the range [0, 1]
>     survivalChance = System.Math.Min(1.0, System.Math.Max(0.0, survivalChance));
>
>     return new BsonDocument
>     {
>         { "Remarks", remarks.Trim() },
>         { "Results", survivalChance },
>         { "IsReadyToCompile", true }
>     };
> }
>
> return MockFunction(Arguments);
> ```

---

> > ### Author Response · Authors · 2024-11-18
> >
> > Following our reproduction guide, users can easily enable Substitution Script and see how the script changes as the training progresses in real-time in the Live Preview Window.

---

> ### Author Response · Authors · 2024-11-18
>
> > "Longer context is not all you need. This analysis should not be drawn from your experiment results. The discovery, however, seems to be interesting, as a conclusion might be drawn from the intrinsic properties of these Kaggle tasks. One or two concrete examples from each experiment should be added to assist the analysis. It should be the properties of the tasks that lead to this phenomenon, not the phenomenon itself."
>
> We agree with your opinions on this phenomenon. In fact, we think it is probably a characteristic of Mockingbird - the current implementation of reflection mechanism also contributed to this phenomenon.
> In section 3.2 (Discussion), we are only discussing about Mockingbird, and the conclusion "Longer context is not all you need" only applies to Mockingbird, rather than the whole field of in-context learning.
> When we write this statement, our actual purpose is to warn our users that increasing context length is not a "silver bullet" that can magically solve all problems - there is a margin utility within it.
> In this sense, our experiment can prove this statement: solely increasing the context length cannot guarantee the improvement of performance in all kinds of tasks.
> We appreciate your suggestions on providing examples and deeply research the reason for this phenomenon. Currently we are collecting more experiment data and analyzing operation logs of previous experiments entry by entry. Thank you for your guidance, and we will immediately report once we found something.
>
> > "From the first paragraph, I can read about the intuition behind compressing and replacing memory, but how to conduct that memory compression is undefined. It would be helpful to provide a detailed description of your implemented memory compression algorithm or to include pseudocode for their approach, as the current description sounds more like a survey and compilation of different methods without a specific implementation."
>
> We apologize that we didn't give enough explanation on memory replacing and compression techniques.
>
> Current implementation of memory replacing is described in section 2.4, line mark 345 (the last sentence of paragraph 2), "We provide a default implementation of the replacing policy
> by replacing correct invocations (without reflection notes) with the latest reflected invocations.".
>
> And current implementation of memory compression is described in section 2.4, line mark 353, "Similar to methods proposed by Wang et al. (2024) and Jiang et al. (2024b), we provide a default implementation of semantic compression (“soft compression”), by instructing LLMs to summarize the reflection notes to compress to previous invocations."
>
> However, you are right, these explanations are too short and abstract to a degree. We didn't make much innovation in this part, so we didn't talk too much about the details in the original paper.
> We will add a pseudo code to prevent potential inconvenience for future readers. Thank you for your advice.

---

> ### Author Response · Authors · 2024-11-18
>
> >"The authors recommend that the user fine-tune self-host LLMs, but it seems they were not responsible for this claim. If all the tasks are binary classification or regression, it would not be difficult to finetune an LLaMA-3 8B model to generate reasonable feedback."
>
> There is an obvious and huge misunderstanding here: we have never recommended users to fine-tune self-host LLMs to generate reasonable feedback.
>
> At the end of section 2.2, the original sentence in the paper is "For users using this paradigm with self-host LLMs, we highly recommend them to **fine-tune their models to schema control instructions**."
>
> The full context is:
> "For LLMs which currently do not support this feature, Mockingbird uses a JSON schema validator to verify the formal correctness of responses, and reject illegal responses with detailed error report to initiate another request-response round.
> For users using this paradigm with self-host LLMs, we highly recommend them to **fine-tune their models to schema control instructions**."
>
> We have never recommended users to fine-tune self-host LLMs to generate reasonable feedback. In fact, (a) Mockingbird does not require users to fine-tune their LLMs to specific tasks; (b) the difficulty of fine-tune LLMs to schema control instructions (JSON schema) does not vary according to different tasks.
>
> Some LLM inference service providers have adapters to enable "structured output" (JSON schema) for LLMs does not initially support this feature. [1]
>
> There are also many tutorials on the Internet about how to fine-tune LLMs to these features on your own. For example, this page is what we just found by googling with keywords "fine tune json schema".[2]
>
> We can understand this misunderstanding but still feel sad about this unfounded accusation of "not responsible".
>
> But still, the existence of this misunderstanding reflects that our expression in this paragraph is still not straightforward enough. We will add more text to instruct users to adapt this paradigm to LLMs without JSON schema feature. Thank you for helping us improve user experience.
>
> [1] LoRAX + Outlines: Better JSON Extraction with Structured Generation and LoRA - Predibase - Predibase (https://predibase.com/blog/lorax-outlines-better-json-extraction-with-structured-generation-and-lora)
>
> [2] imaurer/awesome-llm-json: Resource list for generating JSON using LLMs via function calling, tools, CFG. Libraries, Models, Notebooks, etc. (https://github.com/imaurer/awesome-llm-json)

---

> ### Author Response · Authors · 2024-11-18
>
> >"If all the problems are binary classification and regression containing a single answer, why is JSON schema output necessary? I am not saying there are specific drawbacks, but introducing JSON schema for this simple scenario is confusing. Why don't we just prompt the LLM as a multi-choice QA?"
>
> You are right, in simple scenarios when results are single primitive values (one integer, one boolean, or one string), JSON schema is not necessary.
>
> However, as a library, Mockingbird is designed to serve general purpose functions; and in scenarios of developing modern software in reality, it's common to see functions that take complex types (structures and classes that have multiple fields or properties) and nested types as parameters and return values. Also, in these scenarios, directly prompting LLM is not an efficient solution. Therefore, the use of JSON schema (and the support for arbitrary types as parameters and return value) is not a redundant design, but a necessity for Mockingbird to be helpful for developers.
> After all, Mockingbird is an open-source library, and we wish that it can really help developers in their development.
>
> > "The paper suggests that the higher number of data elds in the "Horses" task contributes to the difficulty. Explain why this might be the case."
> >
> > "The explanation of the "guessing trap" is too abstract. Use concrete examples from the "Horses" task to illustrate how the LLM might be making incorrect assumptions about its errors and how those incorrect assumptions lead to a lack of improvement."
>
> These are indeed the points we have overlooked before. Thank you all for pointing out this.
>
> After reading your comment and also comments from reviewer 4oeR, we realize that this indeed worth an in-depth analysis and the answer could probably further exploit the potential of this paradigm.
>
> We plan to add the "domain-specific knowledge" as a factor into our experiment:
>
> (a) as for our "guessing trap" theory, we plan to seek help from real veterinarians - to see if LLMs can actually start learning after human experts (veterinarians) correcting their initial wrong reasonings or providing valid reflection notes.
>
> (b) we plan to test if retrieval-argumented-generation with domain-specific documents (veterinary articles and books, offline or online) can solve this problem.
>
> If our users may face this problem in the future, then it indeed is our responsibility to provide at least a mitigation solution, thank you for reminding us of this.
> We will report our result as soon as we have completed our experiment on this.
>
> > "More ablation studies regarding each component of Mockingbird are helpful. For example, could using serializers for both input and output boost performance? How and what information should be compressed within the memory compression steps? Or should we use a monkey patch substitution script directly generated by LLM in the training loop?"
>
> Thank you, these suggestions are very helpful. Currently we are conducting non-conventional "ablation studies" - those optional techniques, such as Substitution Script, Memory Replacing and Memory Compression.
>
> They have not been quantitively evaluated in experiments in the paper - we only present quality conclusions on their performance, like "substitution script could improve the response time but reduce the accuracy", which cause many readers wondering about the actual percentage of effects.
>
> Now we are collecting performance data of these optional techniques to show users the quantitative results of these optional techniques' pros and cons, in order to help users select most suitable optional techniques for their own scenarios.

---

> ### Author Response · Authors · 2024-11-24
>
> > "More works adapt LLMs to machine-learning tasks [2, 3, 4, 5], though not all are relevant to Kaggle competitions."
>
> Thank you for providing these related works. We have carefully read these papers.
>
> To briefly summarize these works,
>
> [2] presents a framework that utilize case-based-reasoning LLMs to automatically understand the task and thus generating pipeline code in Python to compose existing conventional machine learning components.
>
> [3] presents a similar framework (compared to [2]) that also produce pipeline code in Python, with a different goal to incorporate domain-specific knowledge into automated machine learning and accomplish automatic feature engineering.
>
> [4] and [5] have designed benchmarks for these code-generation-based automated machine learning workflows.
>
>  Actually, we serve different goals with different approaches.
> These researches are trying to instruct LLMs to write conventional machine learning code to assist data scientists; however, we are trying to provide software developers with a alternative choice when they cannot rely on data scientists nor conventional machine learning methods by instructing LLMs to role-play functions.
>
> We feel that some readers may also confuse researches in this field with our work, so we also add this field into section "Related Work" and explain the difference between our work.
>
> Thank you for helping us clarify this.

---

> ### Author Response · Authors · 2024-11-24
>
> > "The ideology of treating LLM call as a function is not new [1]."
>
> Yes, we agree that "the ideology of treating LLM call as a function is not new".
>
> Research [1] presents a server-side optimization technique for LLMs when the content of some requests depends on previous responses; they introduce a concept called "semantic variable" to describe these dependencies. Semantic variables are marked separately as "outputs" in previous responses and "input" in subsequent requests. This input-output relationship is in fact a form of functions.
>
> As you have pointed out, treating LLM calls as functions is not new. In fact, in the field of software engineering, anything that has an input and an output is usually treated as a function. This is nothing new since the last century.
>
> However, the core idea of our work, "instructing LLMs to role-play functions", is different from "treating LLM call as a function" -- we presented not just a vague idea with our paper, but a paradigm with concrete methodology, system design, and corresponding software artifacts.
>
> Before we started our work, one of the co-authors believed that "There should already be some work on this." After investigation, the answer is a disappointing "no".
> During the work, we also found a possible reason for this -- there are many concrete technical problems awaiting to be solved within the design, and solving  them requires a deep understanding of LLMs and software engineering theories, in addition to a lot of engineering work to implement them.
>
> In conclusion, [1] is an interesting work that provides a useful framework, and the input-output relationship within semantic variables does resemble a function. But our work is also more than presenting a vague idea; part of its contribution, as we believed, is a paradigm with concrete methodology as one of the possible implementations of this vague idea.

---

> > ### Comment · Reviewer_hhzP · 2024-12-01
> >
> > I appreciate the author's efforts during the rebuttal period and have updated the overall assessment accordingly. The manuscript has been significantly improved, and the code implementation is intriguing. However, the scope of the experiments is still limited.
> > I look forward to a stronger submission in the next round, where the author can expand this framework to include more tasks and enhance performance.

---

### Official Review · Reviewer_dFTz · 2024-11-03

**Soundness:** 1
**Presentation:** 1
**Contribution:** 1
**Rating:** 3
**Confidence:** 4

**Summary:**

The paper explores the possibility of using LLMs as mock-function-implementors focusing on machine learning tasks. It pits this idea against conventional machine learning methods and offers advantages such as (a) "intrinsic knowledge for free" (b) flexibility in terms of inputs and (c) potential to use agentic behavior to access external resources based on context. The paper describes the workflow of using the Mockingbird platform which involves users writing a mock function (function signature and docs describing the behavior) and an optional training phase where users can give a training set to the platform, which can use a reflection-style process to curate "few shot examples" for runtime usage. The platform takes care of prompting the LLM with function inputs and deserializing the LLM's outputs while providing schema-adherence (either using the LLM provider's built-in support, or reprompting the LLM if response does not adhere to the defined schema). The platform can also support plugging in different memory compression techniques to manage context length.

The paper evaluates the performance of GPT4o plugged with this platform on a few different Kaggle competitions (for classification and regression tasks) and finds that on 3 of them the platform achieves SOTA or competitive scores (while severely underperforming on some other contests).

**Strengths:**

The idea of using LLMs to implement mock functions is interesting.

**Weaknesses:**

1. The paper introduces a lot of ideas but does not substantiate them leaving the reader confused as to how exactly the idea might be implemented (even conceptually).
    - E.g., the entire bit about branching in mock memory is unexplained -- how are the branches used? Does the user of the mockingbird library/platform have to do something explicitly to use these branches? What purpose do they serve? Are they effective at whatever they are supposed to do (no evaluation)?
    - The idea of substitution script is not evaluated
    - Is reflection really effective? How well does Mockingbird perform when you just feed the training examples in a few-shot manner, without any "remarks" field?

2. Since the paper is trying to position itself as defining a new paradigm, it's important to explore the different design choices which the paper does not do at all. I do not think page limit is an issue because sections 1 and 2 have a lot of overlaps and a bit of rewriting could easily give some space for more thorough evaluation. Similarly, some parts in 3.2 could also be cut down (e.g., "Why cannot LLMs be “a good veterinarian”?").

3. The figures and descriptions are quite confusing.
    - E.g. I do not understand how the arrows work in Figure 1, and what is the "reasoning" component there, why it is separate from the LLM, what is the real-time information from external sources etc.
    - E.g., "Based on the feedback from users or error from software systems, this platform will instruct the LLM to conduct chains of thoughts to reflect on its previous output" -- feedback from users/software systems? How that works is not explained in the paper, as the paper seems to position this platform as taking in a training dataset and using reflection style prompting to build a version of the "Mock Memory", which likely happens not at runtime but in training phase?

4. Some of the claims are not really valid. E.g., " Overall, it achieves very competitive scores " is not accurate when the method achieves 6 and 18 percentiles on two (binary?) classification datasets. "However, there is a paucity of research exploring the integration of LLMs into a broader range of intelligent software systems" is also not true (but that's not too concerning to me as that's not central to the paper).

**Questions:**

My main question to the authors is what are the advantages of using Mockingbird over directly prompting the LLM with examples? Given that LLM providers are already providing schema-adhering-responses as an option, the only two benefits I see are: (a) reflection setup for free (b) slightly more convenient interface because of integrating LLMs via mock functions. Is this sufficiently different than calling LLMs directly that we can call it a new paradigm?

I would in fact be quite happy with a library/framework that would let me write mock functions and let LLMs implement them _as long as_ there are sufficiently useful guarantees provided by the library, more than what I would get by raw prompting the LLM. E.g., automatically improving performance (as the authors are trying to do with reflection, but do not really evaluate it thoroughly), mocking stateful objects (it's unclear if the Mock Memory is meant for this also, or just for storing the few shot examples) etc. But these features would have to be evaluated meaningfully.

---

> ### Author Response · Authors · 2024-11-18
>
> Thank you for reviewing our work!
>
> >"E.g., the entire bit about branching in mock memory is unexplained -- how are the branches used? Does the user of the mockingbird library/platform have to do something explicitly to use these branches? What purpose do they serve? Are they effective at whatever they are supposed to do (no evaluation)?"
>
> It seems that we lacked attention to the difference within knowledge background. We will add more detailed explanation of these terms in the appendix.
> In section 2, we described mock memory as "enhanced chat histories".
> The term "chat histories" is frequently used in the implementation of LLM clients.
> In short, a chat history is a complete list of messages from user and assistant.
> Inference API for LLMs are state-less, so clients have to send the full list of past messages to LLMs, and LLMs will return an assistant message according to this list.
> In the following part of the description for mock memory, we listed major unique features compared to regular chat histories.
> We don't have to evaluate this component, because if it is not effective, we will not get any response from LLMs.
>
> We think it is the term "chat histories" causing this question. We will add detailed explanation to this kind of terms in the appdix. Thank you for raising this question!
>
> >"The idea of substitution script is not evaluated."
>
> This indeed is a problem that many other readers may also be concerned about.
> We are re-organizing this paper to move some engineering details to the appendix, and then we will leave more space for substitution script.
> Thank you for pointing out this.
>
> >  "Is reflection really effective? How well does Mockingbird perform when you just feed the training examples in a few-shot manner, without any "remarks" field?"
> >
> >  "... as the authors are trying to do with reflection, but do not really evaluate it thoroughly ..."
>
> We sincerely apologize that it seems we failed to clearly explain the variable "context length" in the experiment section.
> The variable "context length" refers to the number of examples that a mock function can carry in its context.
> When context length is 0, there is no example in the context, which means that the training process (where reflection is conducted) is skipped.
> Therefore, the first column in table 1 & 2 actually reflect LLMs'  0-shot. performance.
> According to the experiment results, the effectiveness of reflection mechanism varies according to different datasets.
> The biggest improvement brough by reflection is observed in "Mushrooms Classification Dataset" when context length is 60; in this setting, the improvement of accuracy compared to 0-shot (when context length is 0) is about 167% .
> We will add the explanation of "context length" to the revised paper, thank you for raising this question.
>
> > "E.g. I do not understand how the arrows work in Figure 1, and what is the "reasoning" component there, why it is separate from the LLM, what is the real-time information from external sources etc."
>
> Sorry for causing your inconvenience.
> Component "reasoning" here refers to the content within the "remarks" field, it's the result that LLMs output after reason.
> It is our fault not to add more information about technical details about LLM clients.
> Currently, clients can utilize plugins (called as "tools" by OpenAI) to get information from external sources, such as the Bing plugin for Semantic Kernel (a LLM client whose development is led Microsoft).
> Typically, retrieval-argumented-generation (RAG) is implemented as plugins. It is a typical example of "external sources".
> We will add more details for these techniques used in LLM clients in the appendix.

---

> ### Author Response · Authors · 2024-11-18
>
> > "E.g., "Based on the feedback from users or error from software systems, this platform will instruct the LLM to conduct chains of thoughts to reflect on its previous output" -- feedback from users/software systems? How that works is not explained in the paper, as the paper seems to position this platform as taking in a training dataset and using reflection style prompting to build a version of the "Mock Memory", which likely happens not at runtime but in training phase?"
>
> Sorry that it seems that our explanation is still not explicit enough.
> This part is explained in section 2.3 (Learning Through Reflections).
> Also, part of this question is raised due to our lack of explanation for "chat histories" (in Mock Memory part).
>
> Mock function provides a function "void Amend(BsonDocument arguments, BsonValue result, BsonValue remarks)", which will search all messages in the mock memory (chat history) which have the same arguments, and then replace their the results and remarks.
> In our implementation, there is no native boundary between "training" and "evaluating" phases for mock functions; developers can invoke the "Amend" function at anytime.
>
> Phases "training" and "evaluating" are introduced by Mock Trainer; we specially designed its workflow to mimic the ordinary workflow for conventional machine learning, so that we can conduct our experiment in section 3.1.
> As discussed in section 2.3, Mock Trainer will separate the dataset into training data and evaluation data, and in the training stage, Mock Trainer will conduct the reflection process, and use "Amend" function to update Mock Memory; and in evaluating stage, Mock trainer will not conduct reflection process nor update Mock Memory, so that we can get fair and accurate results of performance indicators such as overall accuracy.
>
> However, using Mock Trainer is not mandatory.
> User can use Mock Functions without training by Mock Trainer, or  in a reinforcement learning manner - continuously conducting reflection and amending, when real-time feedback is possible (such as real-time stock price crawled from the Internet in financial machine learning).
>
> We will add more explanations about the technical details in the appendix to avoid future readers suffering from similar inconvenience.

---

> ### Author Response · Authors · 2024-11-18
>
> >"My main question to the authors is what are the advantages of using Mockingbird over directly prompting the LLM with examples? Given that LLM providers are already providing schema-adhering-responses as an option, the only two benefits I see are:  (a) reflection setup for free (b) slightly more convenient interface because of integrating LLMs via mock functions."
> >"Is this sufficiently different than calling LLMs directly that we can call it a new paradigm?"
>
> We fully understand your worries about researcher over-claiming results.
>
> Before answering these questions, we have to clarify that: (a) "Mock Function" is a paradigm. (b) "Mockingbird" is a platform implementing the paradigm "Mock Function".
>
> In the field of software engineering, a "paradigm" is usually considered as a perspective to view and model software; today, some paradigms even only have vague definitions.
> We call mock function "a new paradigm", because: (a) it is a paradigm (meanwhile, directly prompting LLMs is not a paradigm, there is no explicit model for developers to follow, there are no corresponding reusable tools for developers to use). (b) it is new.
> As we can summarize "micro-service paradigm" as "diving a huge system into many small services", "Mock Function" is a paradigm about instructing LLMs to role-play functions.
>
> As for the question of advantages, this question is a little hard to answer, because the advantages of using C++ over directly programming the computer with machine code can also be summarized as  "a slightly more convenient interface", but it is also an true but unfair oversimplification, for "slightly" is subjective and undefined.
>
> In section 2.1, we have discussed the whole workflow of *Mockingbird*, where readers can find that JSON schema is an important feature that it relies on, rather than its substitution - users cannot achieve "instructing LLMs to role-play functions" by solely restricting the responses from LLMs with JSON schema.
> As for human users, they can gain all the benefits that Mockingbird can provide by directly prompting LLMs (obviously they can implement serialization, reflection, and all other techniques by directly prompting), as long as they can endure this boring and meaningless process. (Just like programmers can still write programs with machine code.)
> The cost of using Mocking is to simply click the button to install it through the package manager, so we believe a more fair question to discuss is the disadvantages of not using Mockingbird:
> (a) developers have to spend more time on re-inventing wheels; (b) developers have no community to share reusable tools, or discuss questions, for they are using their own model of prompting LLMs to organize their paradigm.
>
> As for the major advantages of using the paradigm (even though this is not included in your question), as we summarized in section 1, are: (a) the intrinsic knowledge of LLMs acquired from the pre-training data enables this paradigm to perform well on few-shot and zero-shot tasks; (b) this paradigm is not constrained by a strict input data schema, allowing it to process incomplete data entries with missing fields. (c) this paradigm is readily capable of utilizing tools and extracting information from non-structural sources, which are typically inaccessible to conventional machine learning techniques.

---

> > ### Author Response · Authors · 2024-11-18
> >
> > Here are some of brief examples of how applications in reality may benefit from using Mockingbird (or some others' implementation of Mock Function paradigm):
> >
> > (a) intrinsic knowledge: if you are a developer who want to develop an app to help users estimate the prices of their second-hand cars to sell, then you can directly use Mockingbird to only write the method signature for this method and use it, without the whole training process in conventional machine learning methods; as shown in the experiment for "Car Prices Estimation" dataset, some LLMs already contains intrinsic knowledge for this task, and they can get acceptable scores only relying on their intrinsic knowledge and reasoning ability.
> >
> > (b) tolerance for input data: data scientists all know that they have to pre-process missing values before they start using conventional machine learning tasks, and when the input data for evaluation also contains missing values, then they have to use certain methods to replace these missing values; however, in our experiment for "Titanic Survival Prediction" dataset, we didn't preprocess input data for both of training and evaluation - the missing values are sent to LLMs as "null" values, yet LLMs can normally do the prediction with a tolerance for these missing values.
> >
> > (c) ability of using tools: most of today's LLMs have the ability to use tools, developers can use these tools to provide additional information to LLMs, such as the retrieval-augmented-generation. For example, let's assume you are a developer for stock price prediction program, which has to predict the price of a stock in next few minutes, and with this Bing search plugin, your application can easily have a advantage that none of these conventional machine learning methods can easily achieve - the ability to browse the Internet; the prediction of your application can be based on the latest news about the company who is issuing that stock, which will usually have an impact on the stock price.

---

> > ### Author Response · Authors · 2024-11-18
> >
> > Also, as for the field of software engineering, this characteristic of Mock Functions is very important: the usage of Mock Functions are exactly the same to ordinary functions; to applications, systems, and language runtimes, they are the same things.
> >
> > In software engineering, there is a commonly used effective technique to improve the system performance: object pool; a object pool will store objects that are not currently in use, and in following requests, they will return these stored objects rather than create new objects.
> >
> > The problem regarding this object pool is that how many objects should this pool store at most: storing too many objects (which are usually large, and that's also the reason why they need object pools) will consume a lot of memory, and  storing too less objects will make this pool less effective because it still needs to create a lot of new objects for these requests.
> > Currently, the capacity of an object pool is usually a static number, which is not very "intelligent". Obviously this can be a machine learning task, and surely developers can train a neural network to predict the best capacity for current situation, but it is hard to decide which indicators should be selected as input values (requests per minute? available RAM? number of active threads?), and these indicators are usually subject to change according to the process of development; and when new indicators are added as input values, developers have to re-train their neural networks. However, as we have discussed above, due to the advantage (b) of Mock Function, developers can just use a Mock Function to predict the optimal capacity.
> >
> > If we look at implementation details of today's software engineering, we can find many more opportunities like the example mentioned above, where we can bring "the intelligence of LLMs" into these software systems.

---

> ### Author Response · Authors · 2024-11-18
>
> >  "Since the paper is trying to position itself as defining a new paradigm, it's important to explore the different design choices which the paper does not do at all."
>
> We begun this work because we found that there was no suitable options.
> Could you please name some "different design choices" which can solve the problem that Mock Function is trying to solve? So that we will compare Mock Function with them.
> By the way, directly prompting LLMs is not paradigm from the perspective of software engineering.
> Thank you for your help in advance.
>
> > " Overall, it achieves very competitive scores " is not accurate when the method achieves 6 and 18 percentiles on two (binary?) classification datasets.
>
> It seems that we have different understanding of the word "competitive".
>
> As we specially emphasized in the evaluation section (section 3), "It must be emphasized that our aim is to provide a more suitable paradigm for exploiting the capabilities of LLMs in automatic intelligent systems, rather than provide a state-of-the-art method that is comprehensively superior to conventional machine learning methods."
> Mockingbird is submitted to the area of "infrastructure, software libraries, hardware, systems, etc.", also, unlike conventional AI researches which provide methods that can yield higher scores, we propse Mock Function and Mockingbird because we need something like them, and we cannot find similar solutions.
>
> A data scientist or a data analyzer is very likely not to take Mockingbird into consideration, however, as for developers working on fully autonomous systems who are eagering to adapt LLMs into their complex systems (such as us, working on robot systems), they would found these scores listed in the section 3 acceptable and competitive (compared to not using LLMs).
>
> We will add more text to emphasis our goal and explain our motivation. Thank you for reminding us.
>
> > "... mocking stateful objects ..."
>
> It is a good feature proposal. This feature would be more convenient for developers with object-oriented-programming. Since it can be easily implemented by passing a dictionary storing the values of fields and properties to LLMs, we will implement this future in next version of Mockingbird. Thank you!

---

### Official Review · Reviewer_4oeR · 2024-11-04

**Soundness:** 3
**Presentation:** 4
**Contribution:** 4
**Rating:** 10
**Confidence:** 3

**Summary:**

The paper introduces *Mockingbird*, a platform designed to adapt large language models (LLMs) for a variety of general machine learning tasks. The approach leverages LLMs' reasoning and in-context learning abilities through "mock functions," which enable LLMs to role-play predefined functions without needing traditional code implementations. This setup allows for interactive machine learning by transforming LLMs into flexible, general-purpose functions that adapt based on user feedback and system errors.

Mockingbird is distinctive in that it operates mock functions at runtime, bypassing compile-time code generation. Users interact with mock functions as they would with conventional functions, while the system dynamically manages function calls, converts data between formats, and validates outputs. The platform includes modules for "reflection" (learning from errors), "memory compression" (optimizing memory usage), and "substitution scripts" (code generation after adequate training), which enhance the robustness and efficiency of LLM-driven machine learning tasks.

Mockingbird demonstrates notable advantages: strong zero-shot and few-shot performance due to LLMs’ intrinsic knowledge, flexibility in handling incomplete data, and the capability to integrate tools unavailable to traditional methods. Evaluations on various machine learning tasks (e.g., classification, regression) suggest competitive performance, often surpassing human scores on some datasets. However, the paper identifies limitations, including LLMs' difficulties in self-correcting without feedback and challenges with tasks of higher complexity.

Overall, Mockingbird provides a paradigm shift for using LLMs in machine learning, focusing on efficiency, adaptability, and leveraging LLMs’ unique capabilities in broader intelligent systems.

**Strengths:**

The paper presents several notable strengths across originality, quality, clarity, and significance, offering a comprehensive approach to adapting LLMs for general machine learning tasks through the innovative *Mockingbird* platform. Below is an assessment of each dimension.

### Originality
Mockingbird introduces a unique paradigm that leverages LLMs’ capabilities beyond language processing, adapting them to various machine learning tasks by implementing mock functions that operate at runtime. This approach diverges from conventional methods of LLM utilization by bypassing compile-time code generation in favor of dynamic role-playing, which broadens the application potential of LLMs to areas previously inaccessible. The authors' incorporation of *reflection mechanisms*, which allow LLMs to learn from errors and refine responses over time, adds further originality, as this functionality is typically limited in standard machine learning frameworks. The use of LLMs as flexible, adaptive function substitutes demonstrates a creative synthesis of existing concepts, empowering LLMs to perform tasks traditionally assigned to symbolic or rule-based systems.

### Quality
The paper demonstrates a high standard of quality in both design and evaluation. The *Mockingbird* platform is well thought out, with clear specifications for mock functions, mock trainers, memory management, and substitution scripts. These components reflect careful consideration of performance, accuracy, and computational efficiency, addressing the key challenges that come with deploying LLMs in real-world intelligent systems. The authors provide rigorous empirical validation across multiple datasets from Kaggle, comparing *Mockingbird*’s performance against human benchmarks and conventional methods. The results are promising, showing that *Mockingbird* can often outperform human scores and exhibit robust zero-shot and few-shot capabilities, underscoring the quality and practical relevance of the platform.

### Clarity
The paper is clearly written and logically organized, making the technical details accessible and the platform’s contributions easy to understand. Complex concepts such as *mock functions* and the reflection mechanism are explained thoroughly, and the inclusion of high-level workflow diagrams enhances clarity by offering a visual breakdown of the system. Additionally, the authors carefully define the terms and structure of the mock functions, ensuring that readers unfamiliar with this kind of LLM application can grasp the framework and its potential. The clarity extends to the evaluation section, where performance metrics and dataset descriptions are presented in an easily interpretable manner.

### Significance
Mockingbird’s approach has significant implications for the field of machine learning and intelligent systems. By adapting LLMs to function as mock functions for general machine learning tasks, the authors demonstrate a novel way to harness the latent capabilities of LLMs, which can impact fields requiring flexibility, adaptability, and minimal data-dependency. This paradigm could serve as a foundational model for future systems that aim to integrate LLMs across diverse applications, especially where real-time adaptability and rapid learning from limited data are crucial. Furthermore, by proposing a method that requires minimal fine-tuning or data preprocessing, *Mockingbird* positions itself as a viable solution for practical deployment, potentially reducing the resource costs associated with traditional machine learning pipelines.

In summary, *Mockingbird* presents a significant, original, and high-quality contribution to the application of LLMs in machine learning, broadening their scope and utility with a well-designed, clear, and adaptable framework. This platform has the potential to inspire further research and development in leveraging LLMs for non-linguistic tasks across a wide range of intelligent systems.

**Weaknesses:**

While *Mockingbird* is a promising and innovative platform for adapting LLMs to general machine learning tasks, several weaknesses could be addressed to strengthen the work further.

### Limited Exploration of Baseline Comparisons
One notable limitation is the lack of detailed comparison with baseline machine learning models or alternative LLM-driven methods in more depth. While the authors compare *Mockingbird* to human competitors on several Kaggle tasks, a more rigorous analysis could involve benchmarks with established machine learning frameworks or even fine-tuned LLMs on the same tasks. This would not only provide a clearer picture of *Mockingbird*'s performance relative to existing solutions but also highlight specific areas where the platform excels or falls short. Including performance benchmarks against these baselines on more complex, real-world datasets (such as those requiring deeper reasoning or specialized knowledge) would provide actionable insights into *Mockingbird*'s strengths and limitations.

### Scalability and Efficiency Concerns
Although the authors mention *Mockingbird*'s ability to reduce inference costs through substitution scripts, there is limited discussion on the scalability of this platform for larger datasets or environments where memory and computational resources are constrained. For instance, the current reflection and mock memory mechanisms may lead to substantial memory consumption and processing time as the number of invocations grows. Further exploration of memory management techniques, such as more advanced context compression strategies or selective pruning of reflection notes, could enhance scalability, making the platform more viable in low-resource or high-throughput settings. A more comprehensive assessment of time and memory consumption across different LLMs would provide valuable guidance for deploying *Mockingbird* in various real-world contexts.

### Ambiguities in the Reflection Mechanism
While the reflection mechanism adds a unique learning dimension, the paper lacks clarity on how effectively this mechanism generalizes across diverse types of errors and tasks. For example, the platform could struggle with more nuanced errors, particularly in cases where initial invocations are flawed or where errors are complex and multilayered (e.g., multi-step reasoning tasks). Additional experiments that specifically measure how the reflection mechanism handles errors of different complexities and across a range of machine learning tasks could help delineate its robustness. Moreover, exploring alternative self-correction or meta-learning techniques, possibly by incorporating user feedback or additional task-specific criteria, might enhance the flexibility and reliability of the reflection process.

### Limited Domain Exploration and Specificity
The paper evaluates *Mockingbird* on a relatively limited set of datasets from Kaggle, which may not fully capture the platform's potential across varied or specialized domains. This limitation is especially relevant given *Mockingbird*’s stated goal of general applicability. Incorporating datasets from more specialized fields (e.g., biomedical, financial, or scientific data) would provide clearer insights into the platform's effectiveness and adaptability in specific, high-stakes applications. By demonstrating *Mockingbird*'s utility in niche or technically challenging tasks, the authors could better validate the generality and practical impact of their approach.

### Lack of Error Analysis and Interpretability
Currently, the paper lacks a robust error analysis to explain cases where *Mockingbird* performs suboptimally, particularly in tasks like the Horse Health Outcome Prediction. Understanding why the platform fails on certain tasks, potentially due to intrinsic biases or limitations in LLM reasoning, would yield actionable insights for further improvement. Additionally, including an interpretability component could help users understand *Mockingbird*'s predictions and errors, improving trust and usability. The authors could consider incorporating explainability methods (e.g., output sensitivity to input variations or traceability of the reasoning chain in reflection notes), which would help address the challenges users might face in interpreting model outputs.

### Practical Constraints and Implementation Details
While *Mockingbird* presents a flexible and promising framework, the practical challenges of implementing it remain underexplored. Real-world deployment details, such as the costs associated with using commercial LLMs, latency issues, and hardware requirements, are not fully discussed. For instance, clarifying how *Mockingbird* performs under constraints like low latency or limited budgets would add significant value for practitioners. Further discussion on potential optimizations or variations in LLM setups (such as use cases where smaller, more cost-effective models could be used) would also enhance the work’s applicability.

### Actionable Improvements
To address these weaknesses, the authors could:
1. Expand on baseline comparisons with existing ML and LLM-driven methods on complex tasks.
2. Provide an in-depth scalability analysis, especially focusing on memory and inference cost management.
3. Clarify and test the robustness of the reflection mechanism across diverse error types and complex reasoning tasks.
4. Demonstrate the platform's adaptability in specialized and domain-specific datasets.
5. Conduct a detailed error analysis and explore interpretability mechanisms to increase user trust.
6. Address practical constraints related to real-world deployment, potentially offering guidance on optimizing LLM configurations under resource constraints.

**Questions:**

1. **Clarification on Reflection Mechanism Robustness**
   - Could you provide more detail on how the reflection mechanism adapts to different types of errors across varied tasks? Specifically, does the mechanism struggle with more complex or multi-step reasoning errors? Additional information on whether the reflection process performs differently for tasks with high levels of complexity would clarify the flexibility and potential limitations of this component.

2. **Baseline Comparisons with Alternative Machine Learning Models**
   - To better assess *Mockingbird*'s advantages and limitations, could you expand on the rationale for selecting the current benchmarks? Would you consider adding direct comparisons with other machine learning models or frameworks, such as fine-tuned models, to give a fuller picture of the platform’s relative performance? Clarifying this would strengthen the validity of the claims regarding *Mockingbird*'s competitive advantages.

3. **Scalability and Memory Management in High-Throughput Settings**
   - What strategies do you envision for managing memory and scaling *Mockingbird* in high-throughput or resource-constrained environments? Since the reflection and memory mechanisms could become resource-intensive as the number of invocations grows, an explanation of any built-in controls or suggestions for extending *Mockingbird* in real-world applications with limited memory or computational power would be valuable.

4. **Error Analysis and Interpretability**
   - The paper could benefit from a more in-depth error analysis to explain cases of suboptimal performance, such as the Horse Health Outcome Prediction task. Could you provide more details on how intrinsic biases or model limitations might affect *Mockingbird*'s outcomes in these scenarios? Additionally, would you consider incorporating interpretability tools to aid users in understanding and trusting the model's outputs?

5. **Applicability to Specialized Domains**
   - Could you elaborate on how *Mockingbird* would perform in highly specialized domains, such as biomedical or financial datasets, where domain-specific knowledge or more complex reasoning may be required? Testing on such datasets would help validate *Mockingbird*'s versatility and potential for generalization, particularly in fields where accuracy and reliability are critical.

6. **Detailed Analysis of Inference Costs and Deployment Feasibility**
   - Real-world implementation details such as the costs associated with using commercial LLMs, latency issues, and hardware requirements are not fully addressed in the paper. Could you provide an estimate or guidance on the practical costs, especially for larger models in commercial settings? A discussion of potential cost-effective alternatives or guidance on deploying smaller LLM configurations would help practitioners consider *Mockingbird*’s feasibility in production environments.

7. **Plans for Future Work and Domain Expansion**
   - The conclusion suggests potential future directions for *Mockingbird*, particularly regarding real-time LLM inference. Could you elaborate on any specific domains or additional types of machine learning tasks you believe would be particularly well-suited for future implementations of *Mockingbird*? Expanding on this would help readers understand where the paradigm might have the most immediate impact or unique advantages.

---

> ### Author Response · Authors · 2024-11-18
>
> Thank you for reviewing our work and help us improving it!
>
> > "Could you provide more detail on how the reflection mechanism adapts to different types of errors across varied tasks? Specifically, does the mechanism struggle with more complex or multi-step reasoning errors? Additional information on whether the reflection process performs differently for tasks with high levels of complexity would clarify the flexibility and potential limitations of this component."
>
> Thank you for providing the guidance. Current design of Mockingbird serves as an universal and autonomous solution, also we have to admit that there is a lack of optimization techniques for specific errors in our paper. But we agree that the types of errors within reasoning contents may affect final scores, and conducting in-depth analysis on these error types may lead to solutions that can improve its robustness across various domains.
>
> In future research, we will try to summarize common types of reasoning errors and provide our users with the guidance to handle these reasoning errors and thus improve its performance.
>
> >"The paper could benefit from a more in-depth error analysis to explain cases of suboptimal performance, such as the Horse Health Outcome Prediction task."
>
> Thank you for this advice. We are currently conducting an analysis on the reasoning process of operation logs for the Horse Health Outcome Prediction task, and we plan to introduce domain knowledge (providing veterinarian documents for retrieval-augmented-generation or inviting a real veterinarian to fix LLM's reasoning content) into experiments to further enhance our analysis in the Discussion part (section 3.2).
>
> We will report immediately as soon as we have completed this analysis.

---

> ### Author Response · Authors · 2024-11-18
>
> > "Could you provide more details on how intrinsic biases or model limitations might affect *Mockingbird*'s outcomes in these scenarios?"
>
> In Poisonous Mushrooms Classification task, as we have shown in table 1, LLM's scores with insufficient training (when context lengths are 0 and 20) are very low, and when the context is 0 (zero-shot), the score is 0.3720 which is even below the accuracy of random guess.
>
> When there is no training procedure, the LLM only relies on its intrinsic knowledge to perform the reasoning, however, sometimes the intrinsic knowledge that LLM gained from pretraining data is not "true" for its current task.
>
> For example, by comparing the "remarks" field (reasoning) in the operation logs for context length 0 (when accuracy is 0.3720) and context length 80 (when accuracy is 0.8359), we found the LLM's understanding of the correlation between mushroom almond odor and toxicity have a significant change after training:
> in reasonings without training, it believed that almond odor is a sign of poisonous,
> "The mushroom has an almond odor, which is often associated with poisonous species. ..." (Log Item 66fc730b8b1eb660f79c42d8, in "Mushrooms-Context_0-Training_0-37.20%.json")
> however, after training, it believed that almond odor is more often related to edible species,
> "The almond-like odor, yellow cap, and spore print characteristics are commonly associated with edible mushrooms. These features suggest the mushroom is likely safe for consumption."
> (Log Item 66fc944a4e779f389770f298, in "Mushrooms-Context_80-Training_80-83.5870%.json").
> Despite of this difference within the understanding of mushroom odor, we also observed that: after training, it began to consider various factors more comprehensively (such as odor, cap, spore sprint in the example) instead of relying too much on the single factor of odor.
>
> However, we are still unable to quantitively understand how much the actual weight of these properties within LLM's reasoning process have changed (hopefully future researches can provide a tool to do so), but we have qualitative observed that the LLM in use has a intrinsic incorrect preference for relying too much on the single factor of odor, which caused its abnormal score below random guessing (0.5000).

---

> ### Author Response · Authors · 2024-11-18
>
> >"Additionally, would you consider incorporating interpretability tools to aid users in understanding and trusting the model's outputs?"
>
> Currently, users can mainly use operations logs which are reported in real-time to inspect the almost the whole process of training and evaluating, and there is only a UI window of previewing Substitution Script.
>
> However, during our recent analysis, we also realized that more UI tools (especially interactive tools) can be extremely helpful.
> Currently we are designing a UI tool to allow users to inspect the reasoning process, edit them and provide feedback entry by entry in run-time.
>
> We also have plans to introduce LLM-based UI tools to perform semantic analysis on the reasoning content to help users better understand "why LLM thinks so".
>
> > "Could you elaborate on how *Mockingbird* would perform in highly specialized domains, such as biomedical or financial datasets, where domain-specific knowledge or more complex reasoning may be required? Testing on such datasets would help validate *Mockingbird*'s versatility and potential for generalization, particularly in fields where accuracy and reliability are critical."
>
> We have to admit that current implementation of Mockingbird failed to perform well without external domain-specific knowledge, as shown in the Horse Health Outcome Prediction task. Currently we are testing universal applicable solutions such as interactive feedback from human experts in the training procedure and retrieval-augmented-generation with documents.
> Thank you for your advice.
>
> > "Real-world implementation details such as the costs associated with using commercial LLMs, latency issues, and hardware requirements are not fully addressed in the paper. Could you provide an estimate or guidance on the practical costs, especially for larger models in commercial settings? A discussion of potential cost-effective alternatives or guidance on deploying smaller LLM configurations would help practitioners consider Mockingbird's feasibility in production environments."
>
> Sorry that we do lack explanations on its scalability and practical costs. Currently we are conducting money and time cost analysis on the operation logs for previous experiments, scalability experiments on various LLMs, and "ablation experiment" on optional techniques including Substitution Script, Memory Replacing and Memory Compression.
> We agree that these details can help our users better select configurations for their specific tasks, especially when computational resources are limited.
>
> We will report our results as soon as we have completed these experiments and add them to the paper. Thank you for your suggestions.

---

> ### Author Response · Authors · 2024-11-18
>
> > "The conclusion suggests potential future directions for *Mockingbird*, particularly regarding real-time LLM inference. Could you elaborate on any specific domains or additional types of machine learning tasks you believe would be particularly well-suited for future implementations of *Mockingbird*?"
>
> An example that we believe can benefit from this paradigm is the short-term stock price prediction task. An unique advantage of it is utilizing information from non-structural information sources, such as online news websites in this example. By using a mix of the platform and web crawlers, LLMs can make reasonings based on the recent news about the company issuing that stock, thus get a more accurate short-term prediction. In comparison, reports on news websites is hard to utilize in conventional machine learning methods, due to their non-structural characteristic.
>
> Also, from the perspective of software systems, Mock Functions are the same things as ordinary functions, which make them extremely easily to be integrate into existing systems. Currently, many things in the field of software engineering can be improved by machine learning methods, such as deciding the optimal capacity of an object pool. The process of creating a new object instance can be time-consuming, therefore object pools are commonly deployed to store the object instances that are no longer in use anymore, so that next time when an instance is requested, instead of creating a new one, the object pool will return a instance stored in it. However, storing too many objects can consume a lot of memory, while storing too few objects will make object pools less effective because many objects are created due to these unfulfilled requests. Currently developers use a static empirical formula to decide the capacity for each object pools, however, obviously machine learning methods can better predict the number of requests and dynamically scale the capacity, thus reach a better dynamic balance of memory consumption and time consumption, to utilize the memory more efficiently.
>
> In our research field of robot systems, we have an ongoing unpublished project about building API-Independent Application upon Mockingbird. Most of today's applications are API-dependent - developers read the documents about the API that they are about to use and then write the code. API-Independent Application is a reversal pattern - APIs don't tell developers what APIs can do, developers tell APIs what developers need: developers describe the "API" they need as Mock Functions and then distribute these applications as API-Independent Applications. In run-time, these Mock Functions will automatically implement themselves according to the robot systems that they are running on. In this manner, we can develop and distribute applications that can run on robots with different structures (legged, humanoids), which is even beyond visualization to a degree in the past.

---

### Author Response · Authors · 2024-11-13
**Source Code is now Included in the Supplementary Material**

Dear reviewers,

Thank you all for your helpful comments and advice! Currently we are working on these questions and advice, but first we want to share our source code.

We sincerely apologize that we didn't include them in the first version of supplementary material - *Mockingbird* relies on an early version of our unpublished work *Robotless* (whose code is also included in the latest supplementary material). Please note that this version of *Robotless* does not imply its final implementation or performance.

The source code and a reproduction guide can be found under the folder "Source & Documents" in the supplementary material.

Again, thank you all for reviewing!

Sincerely,
Authors

---

### Author Response · Authors · 2024-11-29

Dear reviewers,

Thank you all for helping us improve this paper!

We have reorganized our paper, added more details about the implementation, and conducted several experiments and analyses as requested.

Here is the list of major changes:
- Content
	- Description of Mock Memory is moved to Appendix A.1, more explanation is added.
	- Details about the default implementation of memory replacing and compression is added to Appendix A.2, including pseudo code and prompt template .
	- Content about Substitution Script is moved to Appendix A.4, more details is added, including a graph to explain the workflow and the prompt template.
	- Section 2: the introduction to the workflow of Mockingbird has been streamlined.
	- Section 4:  related works about AutoML are cited and properly compared.
	- Section 3.2:  speculations about the Horse Health Prediction task have been corrected with analysis on the reflection notes.
- Evaluation
	- Appendix B: scalability evaluation including performance, time cost and token usage, and analysis of the reasoning and  reflection process with LLMs with small parameter sizes.
	- Appendix C: evaluation and analysis of the Substitution Script.
	- Appendix D: evaluation and analysis on Mockingbird with retrieval-augmented generation, with examples of how real-world applications can benefit from this paradigm.
- Discussion
	- Appendix E: discussion on the possible limitations imposed by the intrinsic knowledge.
	- Appendix F: discussion on applying Mockingbird to specific domains and corresponding case study on the Horse Health Prediction task.
	- Appendix G: summary on guidelines for deployment Mockingbird in resource constrained environments.
- Supplementary Materials
	- Source code is now included, with a reproduction guide PDF.

---

### Meta-Review · Area_Chair_Vspo · 2024-12-22

**Metareview:**

This paper introduces Mockingbird, a framework that uses LLMs as mock function implementers to handle subsequent general ML tasks. While all reviewers agree that the problem is interesting and the authors have made significant efforts, the paper has received mixed reviews. After reviewing the comments and the rebuttal,  the paper, in its current form, seemingly suffers from several issues, including inadequate presentation of the importance of the mock function and other experimental concerns such as scalability and insufficient comparisons with related work. The authors are encouraged to revise the paper by addressing these issues and providing a clearer justification for the novelty and impact of their approach. With these improvements, the paper could be more competitive for future submissions.

**Additional Comments On Reviewer Discussion:**

While all reviewers agree that the problem is interesting and the authors have made significant efforts, the paper has received mixed reviews. After reviewing the comments and the rebuttal,  the paper, in its current form, seemingly suffers from several issues, including inadequate presentation of the importance of the mock function and other experimental concerns such as scalability and insufficient comparisons with related work.

---

### Decision · Program_Chairs · 2025-01-22

Reject